# Myoglobin-derived iron causes wound enlargement and impaired regeneration in pressure injuries of muscle

Nurul Jannah Mohamed Nasir[1,2], Hans Heemskerk[1,3], Julia Jenkins[1],
Nur Hidayah Hamadee[1], Ralph Bunte[1], Lisa Tucker-Kellogg[1,2,3]*

[1]Cancer & Stem Cell Biology, Duke-NUS Medical School, Singapore, Singapore;
[2]Centre for Computational Biology, Duke-NUS Medical School, Singapore, Singapore;
[3]BioSyM and CAMP Interdisciplinary Research Group, Singapore-MIT Alliance for Research and Technology, CREATE, Singapore, Singapore

**Abstract** The reasons for poor healing of pressure injuries are poorly understood. Vascular ulcers are worsened by extracellular release of hemoglobin, so we examined the impact of myoglobin (Mb) iron in murine muscle pressure injuries (mPI). Tests used Mb-knockout or treatment with deferoxamine iron chelator (DFO). Unlike acute injuries from cardiotoxin, mPI regenerated poorly with a lack of viable immune cells, persistence of dead tissue (necro-slough), and abnormal deposition of iron. However, Mb-knockout or DFO-treated mPI displayed a reversal of the pathology: decreased tissue death, decreased iron deposition, decrease in markers of oxidative damage, and higher numbers of intact immune cells. Subsequently, DFO treatment improved myofiber regeneration and morphology. We conclude that myoglobin iron contributes to tissue death in mPI. Remarkably, a large fraction of muscle death in untreated mPI occurred later than, and was preventable by, DFO treatment, even though treatment started 12 hr after pressure was removed. This demonstrates an opportunity for post-pressure prevention to salvage tissue viability.

*For correspondence:
tuckerNUS@gmail.com

Competing interest: The authors declare that no competing interests exist.

## Editor's evaluation

It is known that muscle pressure injury (mPI), tissue damage caused by sustained pressure, is difficult to be healed although the mechanism underlying has been poorly understood. In this study, the authors demonstrated a convincing evidence that myoglobin (Mb) released at the site of injury plays an important role in the size, severity, oxidative damage, and poor healing of mPI by causing the induction of immune cell (in particular phagocyte) death and delaying the clearance of dead tissues using Mb KO mice and iron chelation by deferoxamine. The authors' findings are valuable in developing a novel therapeutic option for mPI although further clinical corroboration of these findings would be of even greater value.

## Introduction

Pressure injuries (also called pressure ulcers, bedsores, decubiti, or pressure sores) are tissue damage caused by sustained pressure. They are extremely painful (*Girouard et al., 2008*), costly to prevent and treat (*Padula and Delarmente, 2019*), and increase the risk of patient sepsis and death (*Wassel et al., 2020*). Tissue death can be caused by mechanical deformation, ischemia, or both. Ischemia is often studied as ischemia-reperfusion injury, but poor clearance of damage factors may have unappreciated importance (*Gray et al., 2016*).

Pressure injuries often heal poorly (*Mervis and Phillips, 2019*), especially if they involve deeper layers such as muscle (*Bouten et al., 2003*; *Preston et al., 2017*), but the reasons are for poor quality and quantity of healing are not clear. Some cases can be explained by complications or comorbidities (infection, incontinence, poor circulation, hyperglycemia, chronic reinjury, advanced age), but pressure injuries can affect any immobile person (e.g., young adults with spinal cord injury). In this study, we ask whether some aspect of pressure-induced injury is intrinsically inhospitable to regeneration and in need of intervention.

Chronic ulcers of veins or arteries (e.g., venous stasis ulcers, sickle cell ulcers) have high levels of extracellular hemoglobin (Hb) released in the wound. For example, many have deposits of hemosiderin. Extracellular Hb and its breakdown products (e.g., hemin, iron) create oxidative stress (*Goldman et al., 1998*; *Reeder and Wilson, 2005*) and other effects that are detrimental to regeneration. For example, Hb decreases nitric oxide for angiogenesis (*Kato et al., 2017*; *Nader et al., 2020*) and signals as a damage-associated molecular pattern (DAMP) to increase inflammation (*Mendonça et al., 2016*; *Bozza and Jeney, 2020*). Systems exist to detoxify Hb (*Cherayil, 2011*), but Hb is also an innate immune factor with evolutionarily conserved antimicrobial function. When extracellular Hb is activated by proteolytic cleavage, bacterial binding, or conformational change, it increases production of reactive oxygen species (ROS) *Bogdan, 2007*; this process has been called 'oxidative burst without phagocytes' (*Bogdan, 2007*; *Jiang et al., 2007*). Our earlier work used evolutionary conservation to identify ROS-producing fragments of Hb and crosstalk with tissue factor coagulation (*Bahl et al., 2014*; *Bahl et al., 2011*). Therefore, globin proteins have multiple functions that may be detrimental to chronic wounds (*Tchanque-Fossuo et al., 2017*).

Myoglobin release into plasma or urine has been observed after muscle pressure injuries (mPI) in multiple studies including deep tissue injury (DTI) (*Traa et al., 2019*; *Makhsous et al., 2010*; *Loerakker et al., 2012*; *Levine, 1993*; *Traa, 2019*), but was studied as a readout of damage rather than a source of damage. We reason by analogy to hemoglobin that extracellular myoglobin might create a hostile wound environment. An extracellular environment oxidizes globins to a ferric ($Fe^{3+}$) state, which can be further oxidized to ferryl ($Fe^{IV}=O$) globin in the presence of endogenous peroxides such as hydrogen peroxide (*Reeder et al., 2008*). Hydrogen peroxide is ubiquitous in contexts of cell stress, mitochondrial permeabilization, and cell death (*Rojkind et al., 2002*). Ferryl-Mb can oxidize macromolecules directly (*Goldman et al., 1998*; *Kapralov et al., 2009*; *Plotnikov et al., 2009*) and can form heme-to-protein crosslinks (*Osawa and Williams, 1996*). Most importantly, ferryl myoglobin can participate in a catalytic cycle of pseudo-peroxidase activity (redox cycling) (*Boutaud and Roberts, 2011*). In a tissue context, myoglobin can induce ferroptosis, which is a form of non-apoptotic cell death associated with iron and characterized by lipid peroxidation (*Dixon et al., 2012*). Dissociation of myoglobin into free heme or iron results in additional forms of toxicity, as described for hemoglobin.

We hypothesize that mPI will have Mb-dependent pathologies, and that introducing Mb-knockout or iron chelation therapy will partially normalize the mPI pathologies. Deferoxamine (DFO), also known as desferrioxamine or desferoxamine, is an FDA-approved small-molecule iron chelator that improves iron overload syndromes (*Velasquez and Wray, 2022*; *Karnon et al., 2012*). DFO binds free iron and heme at a 1-to-1 ratio (*Velasquez and Wray, 2022*), scavenges free radicals (*Reeder et al., 2008*; *Morel et al., 1992*), reduces ferryl myoglobin to ferric myoglobin (*Plotnikov et al., 2009*; *Vanek and Kozhli, 2022*), inhibits crosslinking of heme to protein (*Reeder and Wilson, 2005*), and prevents the formation of pro-oxidant globin and heme species. DFO can function as an activator of Hif1α (*Tchanque-Fossuo et al., 2017*; *Xiao et al., 2013*), a tool for promoting angiogenesis (*Duscher et al., 2015*; *Holden and Nair, 2019*), an antioxidant (*Sundin et al., 2000*), or can join an anti-ischemic cocktail (*Soloniuk et al., 1992*). DFO appears in hundreds of studies of ischemic or inflammatory pathologies. In our study, subcutaneous DFO is used for testing the hypothesized role of myoglobin iron and as an anti-DAMP therapy for combating local iron overload.

Assessing the contribution of myoglobin iron to pressure injury pathophysiology provides an opportunity to test several additional hypotheses about pressure injuries. First, our prior work in mathematical modeling (*Jagannathan and Tucker-Kellogg, 2016*) predicted that oxidative stress from myoglobin and other DAMPs could create *secondary progression* of pressure ulcers. Secondary progression means that otherwise viable tissue dies later from the environmental consequences of injury, rather than dying directly from the original injury. Pressure injuries are known to have gradual

expansion of tissue death (*Stadler et al., 2004*), consistent with secondary progression, but blocking secondary progression has not been clinically recognized as a goal for intervention (*European Pressure Ulcer Advisory Panel et al., 2019*). Therefore, our studies are designed to test whether tissue margins can be saved from dying, if we initiate iron chelation therapy 12 hr after pressure has ended. Second, we hypothesize that iron chelation therapy, by improving the early stages of injury response, will lead to better muscle tissue architecture (better morphogenesis) in long-term regeneration, even after treatment has ended. This hypothesis will be tested by breeding inducible fluorescence into satellite cells (muscle stem cells bearing Pax7); this fluorescence causes newly regenerated muscle fibers light up against the dark background of pre-existing muscle. Third, establishing a pressure injury mouse model provides an opportunity to learn how much of the poor healing is independent of comorbidities and complications. To the best of our knowledge, pressure injuries have never been assessed for poor regeneration under aseptic conditions in young, healthy animals. We hypothesize that even under these ideal circumstances mPI will heal slowly and incompletely. Our fourth and final additional hypothesis is inspired by prior studies of blood-related conditions in which high levels of hemoglobin or heme/hemin could impair the survival, chemotaxis, and phagocytosis (*Nader et al., 2020*; *Ferris and Harding, 2019*; *Sindrilaru et al., 2011*; *Ballart et al., 1986*; *Liu et al., 2019*; *Martins et al., 2016*; *Chen et al., 2009*; *Yefimova et al., 2002*) of phagocytic cells. Given that pressure ulcers often have slough or eschar, we hypothesize that necrotic tissue will persist in the mPI wound bed, and that sterile mPI will have slough, despite the absence of bacterial biofilm. If correct, this would imply that slough by itself is not sufficient to indicate infection (or bacterial colonization) of a wound.

## Results

### Magnet-induced pressure injury causes delayed healing and failure of muscle regeneration

To compare wound healing between acute and chronic wounds, we injured the dorsal skinfold of mice using either cardiotoxin (CTX) or pressure (*Figure 1—figure supplement 1A–C*) in healthy young adult mice under specific pathogen-free conditions. Both groups of mice received sham treatment (injected with 0.9% saline subcutaneously for 16 d or until mouse sacrifice, whichever was sooner). The normal uninjured mouse skinfold contains the following parallel layers: a thin epithelium (epidermis), a thicker layer of dermis, dermal white adipose tissue (dWAT), a muscle layer called the panniculus carnosus (PC), and a wavy layer of loose areolar tissue (*Figure 1—figure supplement 1D*). Despite comparable diameters of dead muscle between CTX and mPI at day 3, the wound diameters were vastly different at day 10 (*Supplementary file 1a*; p=0.578 at day 3 and p<0.0001 at day 10). In the CTX injury at day 3, many blood vessels were intact and carrying red blood cells, but in mPI at day 3, intact vasculature was not observed in the compressed region (*Figure 1—figure supplement 2*). Prior work showed that pressure affects blood and lymphatic vessels in many ways including potentially loss of flow (*Gray et al., 2016*; *Kimura et al., 2020*; *Karahan et al., 2018*). After CTX killed the panniculus muscle, substantial muscle regeneration occurred by 10 d, in which immature muscle fibers displayed central nuclei (*Figure 1A*). At 40 d after acute injury, muscle was completely regenerated and mature (evidenced by peripherally located nuclei, *Figure 1C*). In contrast, the pressure-injured wound bed remained filled with dead tissue at day 10 (*Figure 1B and E–F*; p<0.0001). Our pressure injuries showed no signs of infection and no epibole (*Figure 1E*). The dead epidermis, dermis, dWAT, and panniculus layers were pushed upward at day 7 ± 2 as slough (necroslough, per the nomenclature of *Nasser, 2019*) and remained at the surface, eventually becoming a dry eschar (*Figure 1E*). When the eschar dropped off (by day 15), the size of the epithelial opening was smaller than the eschar, meaning that re-epithelialization had occurred underneath the eschar. Re-epithelialization completed at day 21 ± 2. Despite successful closure of the epithelial layer, pressure injuries at 40 d had only partial regeneration of the panniculus carnosus layer (*Figure 1D and G*; p<0.0001). At 90 d after pressure injury, the dermis and epidermis had regenerated, but a hole remained in the panniculus muscle layer (*Figure 1H*), indicating a failure to regenerate (*Nasir et al., 2022*). *Supplementary file 1b* summarizes the timelines. We conclude that the mPI healed poorly.

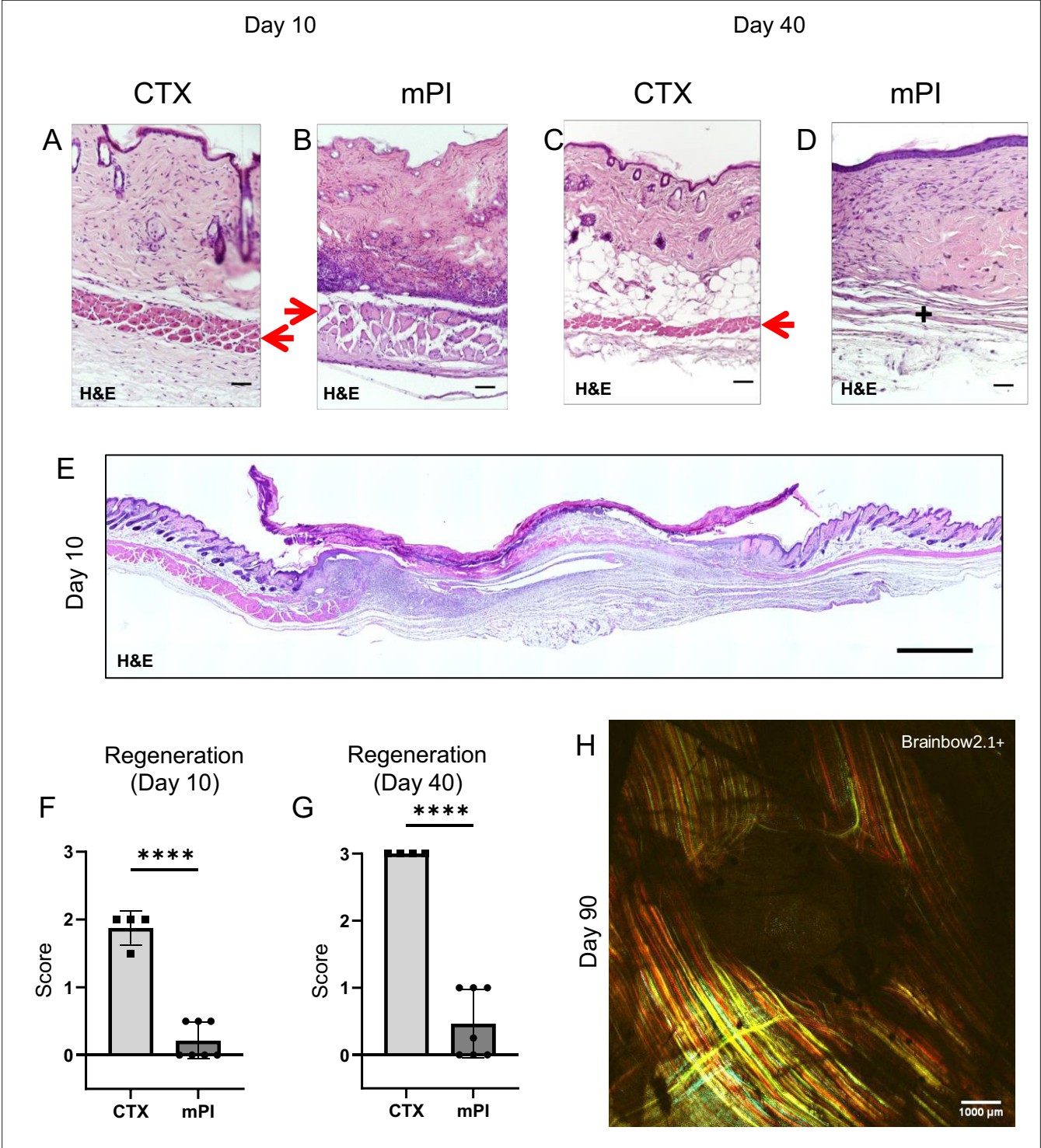

**Figure 1.** Poor regeneration of muscle pressure injury (mPI), a model of chronic wound, compared with cardiotoxin (CTX), a model of acute injury. H&E-stained sections of saline-treated wound tissues (**A, B**) at day 10 post-injury and (**C, D**) day 40 post-injury. Red arrows point to the panniculus carnosus layer. '+' indicates where the panniculus layer should be. Scale bars: 50 µm. Uninjured control tissue, stained with H&E, is shown in *Supplementary file 1C*. (**E**) Full cross-section of H&E-stained mPI, including uninjured edges, at day 10 post-injury. Note the eschar attached to the wound surface. Scale bar: 500 µm. (**F, G**) Comparison of the regeneration scores for the panniculus layer between mPI and CTX injuries at days 10 and 40. (**H**) A multi-channel confocal image of the panniculus layer shows a round hole remains in the muscle, 90 d after mPI. Scale bar: 1000 µm. All injuries are saline-treated to permit comparison against later mPI treatments. All quantitative data are reported as means ± SD. n = 4–7. ****<0.0001 Statistical significance in (**F**) and (**G**) was computed by unpaired Student's *t* test.

*Figure 1 continued on next page*

*Figure 1 continued*

The online version of this article includes the following figure supplement(s) for figure 1:

**Figure supplement 1.** Mouse model of injury.

**Figure supplement 2.** Intact blood vessels are absent or infrequent in the dead tissue of muscle pressure injuries (mPI).

## Compressed regions of pressure injury display an absence of viable immune cells

To investigate why dead tissue remains at day 10, we studied tissue sections from day 3 post-injury. Both CTX-induced and pressure-induced injuries had comprehensive death of muscle tissue, as indicated by karyolysis (dissolution of nuclear components), karyorrhexis (fragmentation of the nucleus), and acidification (eosinification) in H&E staining of the panniculus carnosus (*Figure 2B*). CTX and pressure injuries had a difference in morphology: cells in pressure-injured tissues were flattened, and the thickness of the panniculus muscle layer was half of uninjured (*Figure 2A–C*; p<0.0001). Even more striking was the difference in immune cell numbers. The muscle layer of CTX wounds had sixfold higher levels of immune cell infiltrate than mPI (p<0.0001; *Figure 2A–B and D*). The panniculus layer of mPI was nearly devoid of intact immune cells. Some viable immune cells were found at the margins of the wound (at the boundary between injured and uninjured tissue), but not in the compressed region of mPI (*Figure 2—figure supplement 1*). The absence of immune cell infiltrate is noteworthy because iron-scavenging is performed by macrophages (*Kristiansen et al., 2001*; *Soares and Hamza, 2016*), and because free iron, when not adequately scavenged, can overstimulate the innate immune response (*Bessman et al., 2020*).

Another difference between mPI and CTX was in the level of citrullinated histone-3 (citH3), a marker of extracellular traps (ETs). ETs are formed when phagocytes citrullinate their histones and eject nuclear and/or mitochondrial DNA (and associated factors), which can trap and kill pathogens during infection. The process, ETosis, often kills the host cell. ETs have been observed in sickle cell ulcers (*Chen et al., 2014*). In day 3 wounds, levels of citH3 were tenfold higher in mPI than in CTX (*Figure 2E–G*; p=0.0280). Highest levels occurred near the muscle layer, such as the interface between the panniculus layer and the dWAT or dermis. This is consistent with the possibility that immune cells may have been present in the muscle layer of mPI before day 3 and then died of ETosis. Oxidative stress is a well-studied trigger of ETosis, and other stimuli include hemin and heme-activated platelets (*Okubo et al., 2018*; *Ohbuchi et al., 2017*).

## Free iron remains in wound tissues after pressure injury

Iron deposition was very high in mPI, as measured by Perls' Prussian blue stain (*Figure 2I*), but iron was undetectable at the same time point after CTX injury (*Figure 2H–J*; p=0.0332). Prussian blue detects accumulation of ferric $Fe^{3+}$, typically in the form of ferritin and hemosiderin, but it only shows high levels of iron, because it is unable 'to detect iron except in massive deposition' (*Perl and Good, 1992*; *Liu et al., 2014*). The blue speckles in *Figure 2I* are concentrated iron deposits in the extracellular matrix, and the blue ovals are iron-loaded immune cells (*Kristiansen et al., 2001*; *Soares and Hamza, 2016*). Levels of myoglobin and hemoglobin were also elevated in mPI compared to CTX injury (*Figure 2—figure supplement 2*; p=0.0091 for myoglobin and p=0.0474 for hemoglobin).

Heme oxygenase-1 (HO-1) is an enzyme that performs heme degradation and serves as a marker of high heme or iron. HO-1 was expressed by mPI wound tissues, at similar levels to CTX-injured tissue (*Figure 2K–M*; p=0.998). However, HO-1 expression was localized to the panniculus layer after CTX injury, but it was widespread across all layers after mPI (*Figure 2K–L*). Levels of hemopexin and haptoglobin (iron detoxification factors) are shown in *Figure 2—figure supplement 2*.

## Myoglobin knockout mPI have less tissue death and greater immune infiltrate

To measure the contribution of myoglobin iron to mPI pathogenesis, we developed myoglobin knockout mice ($Mb^{-/-}$) via CRISPR deletion of the entire gene from the germline. Note that prior studies of adult $Mb^{-/-}$ mice found no obvious phenotype (*Garry et al., 1998*; *Gödecke et al., 1999*; *Meeson et al., 2001*). $Mb^{-/-}$ is often lethal to cardiac development during E9.5-E10.5, but some $Mb^{-/-}$ embryos survive to term (*Meeson et al., 2001*). Among our $Mb^{-/-}$ that were born, all developed

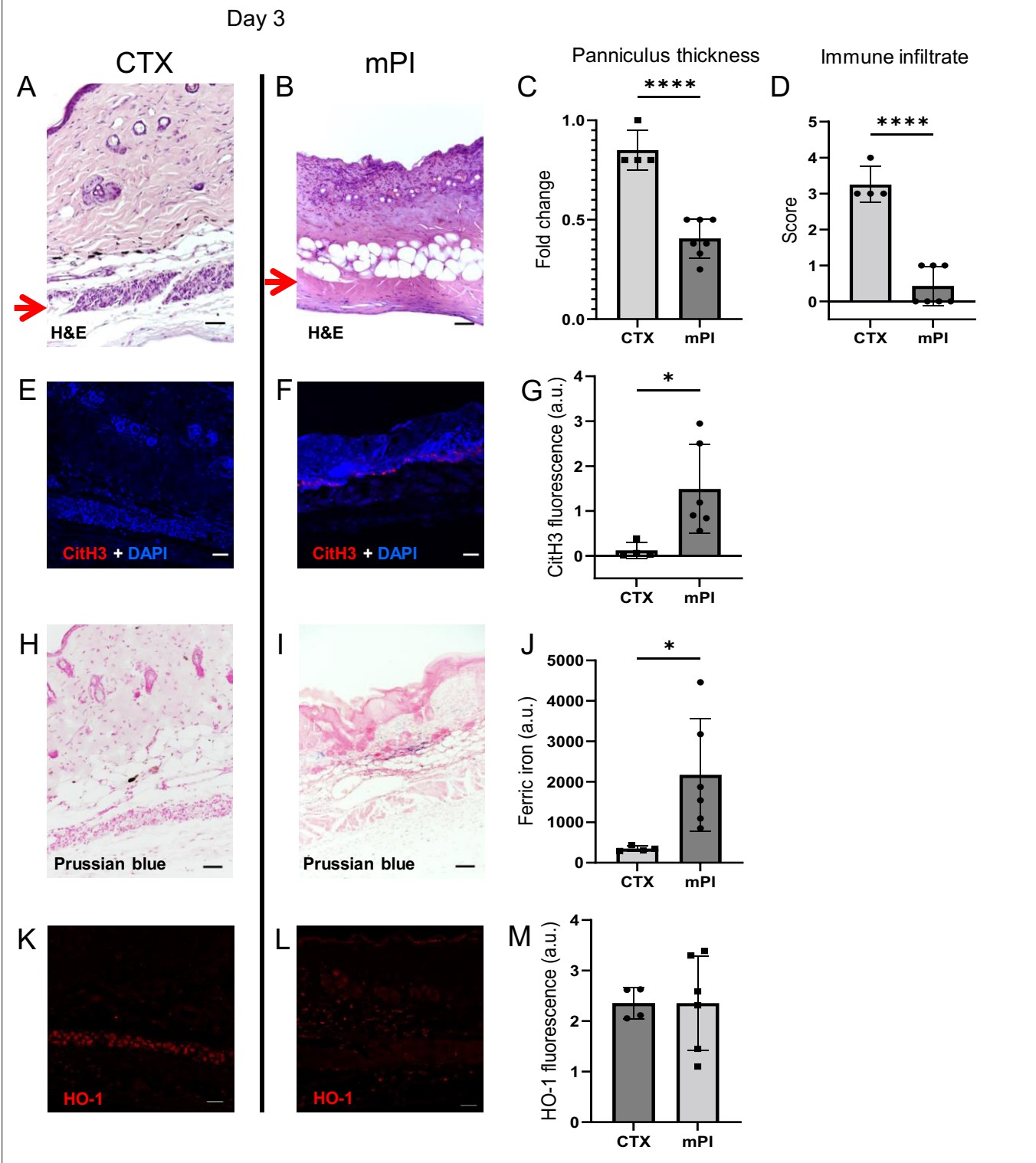

**Figure 2.** Early-stage pathologies in the injury-response of muscle pressure injury (mPI) (3 d after injury). (**A, B**) H&E-stained sections of saline-treated wound tissues, comparing magnet-induced mPI versus cardiotoxin (CTX) injury. Red arrows point to the panniculus carnosus (PC) layer. (**C**) Thickness of PC layer. (**D**) Histopathology scoring of immune infiltrate into the injured PC layer. (**E, F**) Immunostaining for citrullinated histone-3 (CitH3, a marker for extracellular traps, in red) in mPI versus CTX. DNA/nuclei were co-stained blue with DAPI. (**G**) Quantification of CitH3 staining. (**H, I**) Perls' Prussian blue iron staining (dark blue-gray), with nuclear fast red co-stain. (**J**) Quantification of Perls' staining. (**K, L**) Immunostaining for heme oxygenase-1 (HO-1, a marker of heme and iron) in mPI versus CTX. Note that the HO-1-positive signal in CTX injury is localized to the panniculus carnosus and is widespread

*Figure 2 continued on next page*

*Figure 2 continued*

across all layers in mPI. (**M**) Quantification of HO-1 staining. Scale bars: 50 μm. All quantitative data are reported as means ± SD. n = 4–7 mice. *<0.05, ****<0.0001. Statistical significance in (**C, D**), (**G**), (**J**), and (**M**) was computed by unpaired Student's *t* test.

The online version of this article includes the following figure supplement(s) for figure 2:

**Figure supplement 1.** Wound margins show intact immune cells but the compressed region of muscle pressure injury (mPI) lack viable immune infiltrate.

**Figure supplement 2.** Levels of globin-proteins and iron detoxification factors 3 d after cardiotoxin (CTX) injury and muscle pressure injury (mPI).

with normal feeding, weight gain, grooming, social behavior, and life span. Deletion of *Mb* was confirmed by western blotting (***Figure 3—figure supplement 1A***), immunostaining (***Figure 3—figure supplement 1B and C***), and DNA gel electrophoresis (***Figure 3—figure supplement 1D***). With H&E staining, we detected no knockout-induced changes to the tissues of our injury model (skin, panniculus carnosus layer, or loose areolar tissue) other than increased capillary density (by 17%, p=0.0455; ***Figure 3—figure supplement 1E***) and increased thickness of the dWAT layer in *Mb*$^{-/-}$ mice (by 43%, p=0.0232, ***Figure 3—figure supplement 1E***). Total iron was not significantly decreased (p=0.0664, ***Figure 3—figure supplement 1F***).

We compared pressure ulcers in *Mb*$^{-/-}$ versus *Mb*$^{+/+}$ mice (***Figure 3—figure supplement 1G***) using elderly 20-month-old animals. (The mPI in elderly were similar to mPI in young, except with milder increases in pressure-induced oxidative damage, ***Figure 3—figure supplement 2***.) At day 3 post-injury, *Mb*$^{+/+}$ mice had high levels of iron (***Figure 3A***), which appeared in the muscle, dWAT, and dermis, including both the extracellular space and in the small numbers of infiltrating immune cells. In contrast, *Mb*$^{-/-}$ mice had no detectable signal from Perls' stain in any layer of the wound (***Figure 3B and C***; p=0.0020). HO-1 was also decreased by 57% in *Mb*$^{-/-}$ mPI compared to *Mb*$^{+/+}$ (***Figure 3D–F***; p=0.0438). Levels of immune cell infiltrate were 233% greater in *Mb*$^{-/-}$ compared to *Mb*$^{+/+}$ (***Figure 3G–I***; p=0.0250).

The wound size was smaller in *Mb*$^{-/-}$ versus *Mb*$^{+/+}$ (p=0.0297 at day 3, measured as external area, ***Figure 3J*** and ***Supplementary file 1c***). Histopathology scoring in the center of the wound showed 50% decreased tissue death (p=0.0016, ***Figure 3G, H and K***). Oxidative damage was lower in Mb-knockout wounds: DNA damage (8-hydroxy-2'-deoxyguanosine [8-OG]) was decreased by 87% (p=0.0485; ***Figure 4A–E***), and lipid peroxidation (measured using BODIPY 581/591) was decreased by 61% (p=0.0298; ***Figure 4F–J***). BODIPY 581/591 is a fatty acid analogue with specific sensitivity to oxidation (***Drummen et al., 2002***). When oxidized, its emission fluorescence shifts from 595 nm (red) to a maximal emission at 520 nm (green). The green fluorescence is what is shown in the figure. Similarly, *Mb*$^{-/-}$ had roughly 56% decrease in CitH3 (p=0.0127; ***Figure 4K–Q***). These improvements in the wound microenvironment extended beyond the muscle layer because *Mb*$^{+/+}$ wounds had high levels of BODIPY in the dWAT and dermis, and high levels of CitH3 throughout the wound. An additional measure of oxidative damage, 3-nitrotyrosine, showed the same pattern (56% decrease, p=0.0328; ***Figure 4—figure supplement 1***). In summary, the wounds had less oxidative stress when myoglobin was absent. We also measured the impact of Mb on a monocyte/macrophage cell line in vitro. After pre-incubation with pro-inflammatory stimuli, RAW264.7 cells showed an increase in ROS and decrease in phagocytosis (measured as impaired efferocytosis or clearance of dead cells) in response to Mb treatment (***Figure 4—figure supplement 2***). A panel of cytokines, chemokines, and growth factors were measured in muscle homogenates of mPI from *Mb*$^{-/-}$ versus *Mb*$^{+/+}$ at day 3 (***Supplementary file 1d***). There were no significant differences except the knockout wounds had lower levels of CXCL16 (a cytokine associated with lipid peroxidation ***Ma et al., 2018***, ***Supplementary file 1d***; p=0.0110) and higher levels of PAI-1 (also called Serpin E1, a protease inhibitor associated with TGFβ, ***Supplementary file 1d***; p=0.0080). There were no significant differences in total protein between *Mb*$^{-/-}$ and *Mb*$^{+/+}$.

We next sought an orthogonal intervention to test the causal role of myoglobin iron. The FDA-approved iron chelation drug DFO was administered via injection under the dorsal skinfold of 5-month-old mice, starting the morning after pressure induction finished, and repeated twice daily for up 16 d (***Figure 5A***). DFO- or saline-treated tissues were analyzed at 3, 7, 10, 40, and 90 d (***Supplementary file 1e***). The same cohort of saline-treated mPI were compared against saline-treated CTX

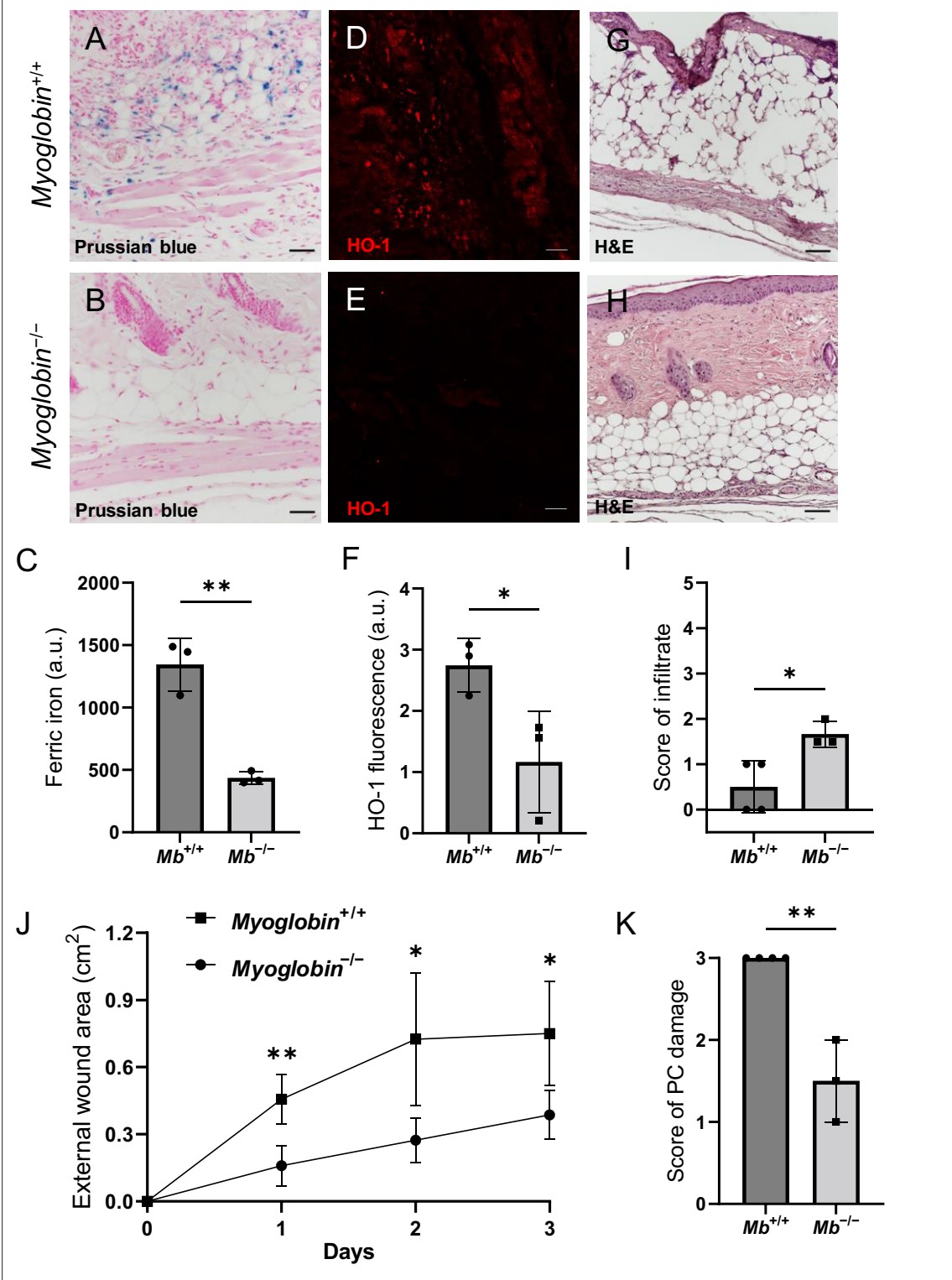

**Figure 3.** Myoglobin knockout decreased iron deposits and tissue death after muscle pressure injury (mPI). (**A, B**) Perls' Prussian blue iron staining. (**A**) Note that iron deposits in the extracellular space and in immune cells of $Mb^{+/+}$ wound tissue, and note that (**B**) $Mb^{-/-}$ tissues have no iron deposits in extracellular regions. (**C**) Quantification of Perls' staining. (**D, E**) Immunostaining for HO-1 in $Mb^{+/+}$ versus $Mb^{-/-}$ tissues at day 3 after mPI. Note that HO-1 is elevated in all layers of $Mb^{-/-}$ except epidermis. (**F**) Quantification of HO-1 immunostaining. (**G, H**) H&E-stained sections of $Mb^{+/+}$ versus $Mb^{-/-}$ mPI. Paraffin-embedded wound sections derived from elderly Mb-wildtype mice had poor cohesiveness (compared to elderly Mb-knockout or young) and exhibited greater cracking during identical sample handling. (**I**) Amount of immune infiltrate, quantified by histopathology scoring on a scale of 0–5,

*Figure 3 continued on next page*

*Figure 3 continued*

performed on day 3 sections. (**J**) External wound area in $Mb^{+/+}$ and $Mb^{-/-}$ mice in the initial days following mPI from 12 mm magnets. Statistical analysis compared four wounds from two age- and sex-matched animals using a Student's *t* test for each day. Consistent with these results, ***Supplementary file 1C*** shows additional animals treated with different-sized magnets. (**K**) Tissue death in the PC muscle layer by histopathology scoring (3 indicates pervasive death). All quantitative data are reported as means ± SD. n = 3 mice. *<0.05, **<0.01. Statistical significance in (**C**), (**F**), and (**I–K**) was computed by unpaired Student's *t* test.

The online version of this article includes the following source data and figure supplement(s) for figure 3:

**Figure supplement 1.** Validation of myoglobin-knockout mice.

**Figure supplement 1—source data 1.** Source data for *Figure 3—figure supplement 1* (western blot).

**Figure supplement 1—source data 2.** Source data for *Figure 3—figure supplement 1* (PCR gel).

**Figure supplement 2.** Comparison of muscle pressure injury (mPI) in young and elderly mice.

in *Figures 1 and 2* (for days 3, 10, and 40 post-injury) and compared against DFO-treated mPI in *Figures 5 and 6*.

## Effects of iron chelation therapy on secondary progression of the wound

Remarkably, DFO treatment caused a decrease in the amount of muscle tissue that died from the initial pressure injury, even though the pressure inductions were identical, and treatments did not begin until 12 hr after the last cycle of pressure. That is, intervention only started after deformation injury and reperfusion injury had already occurred. At day 3 of treatment, DFO-treated wounds had 35% smaller diameter of dead PC tissue (*Figure 5B*; p=0.0043). This observation of secondary progression confirms a prediction of our prior computational modeling (*Jagannathan and Tucker-Kellogg, 2016*). We also observed a decrease in apoptotic debris (measured as TUNEL staining, *Figure 5—figure supplement 1*; p=0.0105), which might reflect a decrease in apoptotic death and/or an increase in phagocytosis/efferocytosis.

## Effects of iron chelation therapy on injury response at day 3

Consistent with decreased death, DFO-treated mPI displayed 77% lower levels of iron deposition by Perls' Prussian blue (*Figure 5C and D*; p=0.0230) compared to saline control (*Figure 2I*) at day 3. As in the elderly mPI, the young saline-treated mPI had iron accumulations in the extracellular space and immune cells of multiple layers, not just the panniculus muscle layer (*Figure 2I*). Similarly, levels of HO-1 were 65% decreased (*Figure 5E and F* and *Figure 2L*; p=0.0071). To confirm that wounds were properly induced, *Figure 5G–I* shows complete tissue death at the centers of the wounds in all animals. Re-epithelialization was not affected by the subcutaneous drug (*Figure 5J* and *Figure 5—figure supplement 2*).

DFO-treated wounds displayed less oxidative damage, as indicated by a 41% decrease in 8-OG (p=0.0048; *Figure 6A–E*) and 50% decrease in BODIPY 581/591 (p=0.0004; *Figure 6F–J*) at day 3. Change in tyrosine nitration was not significant (p=0.965; *Figure 6—figure supplement 1*). CitH3 dropped to undetectable levels in DFO-treated sections (p=0.0105; *Figure 6K–N* and *Figure 2F*). The high CitH3 found in control wounds was partially co-localized with F4/80, a pan-macrophage marker (*Figure 6—figure supplement 2*), but these markers may reflect debris or noncellular localizations. Myeloperoxidase, a marker of neutrophil extracellular traps (NETs/NETosis), was significantly higher in saline-treated mPI compared to CTX (p=0.0173), but showed similar levels to DFO-treated mPI (p=0.946; *Figure 6—figure supplement 3*). *Supplementary file 1f* shows additional measurements of cytokines, chemokines, and growth factors in DFO- versus saline-treated mPI at day 3, and *Supplementary file 1g* shows the same analytes for day 10. These analytes showed no significant differences between DFO and saline treatments, except for CXCL16 levels, which decreased by 36% (p=0.0174) at day 3. Other differences between saline- and DFO-treatment are shown in *Figure 6—figure supplement 3*.

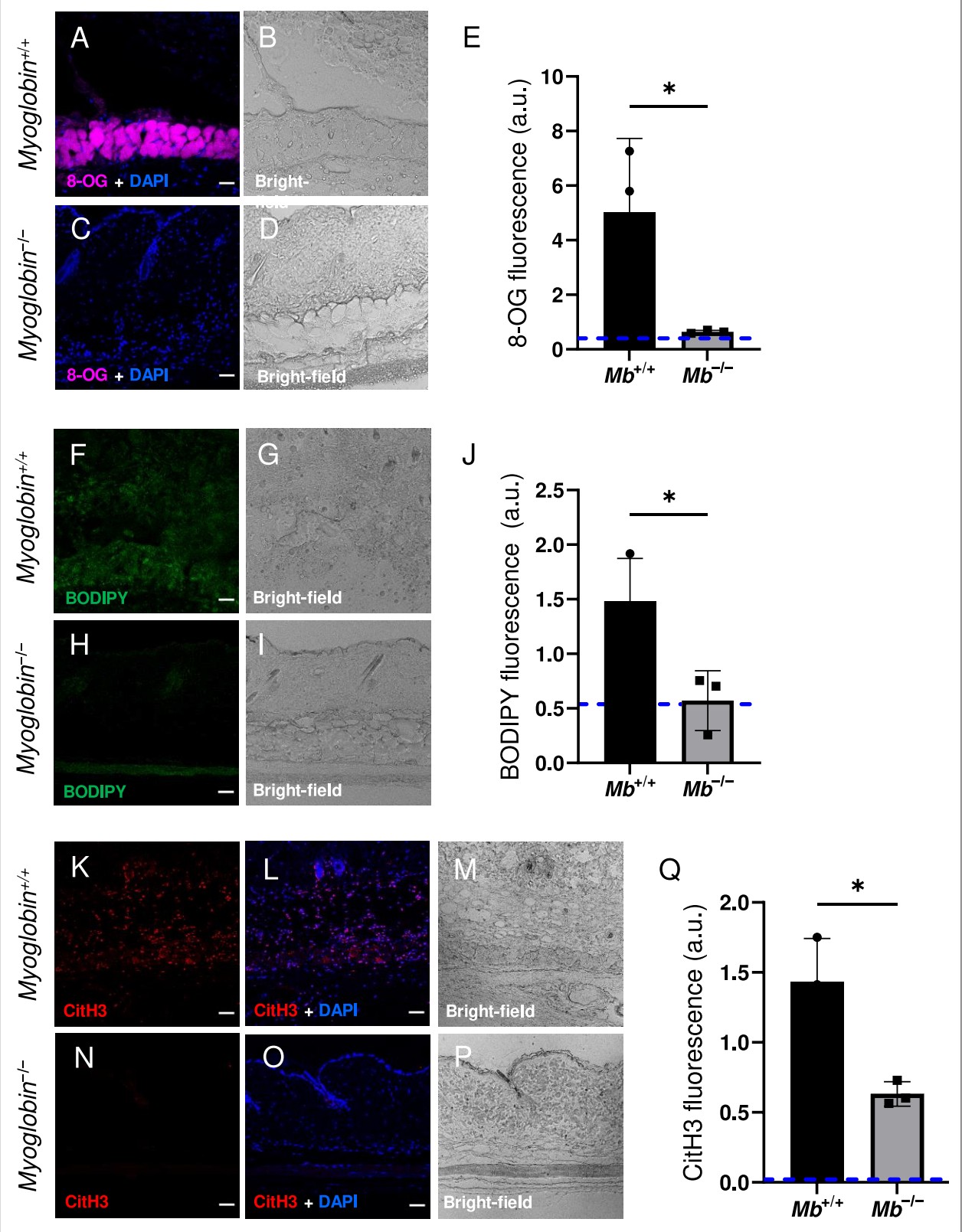

**Figure 4.** Myoglobin knockout caused a more hospitable wound environment in muscle pressure injury (mPI). (**A–D**) Immunostaining of 8- oxaguanine (8-OG; in magenta) in *Mb*$^{+/+}$ versus *Mb*$^{-/-}$ mPI. Nuclei were co-stained blue with DAPI. (**B**) and (**D**) are brightfield images of (**A**) and (**C**), respectively. (**E**) Quantification of 8-OG staining. (**F–I**) BODIPY staining (for lipid peroxidation) in *Mb*$^{+/+}$ versus *Mb*$^{-/-}$. (**G**) and (**I**) are brightfield images of (**F**) and (**H**), respectively. (**J**) Quantification of BODIPY staining. (**K–P**) Immunostaining for CitH3 (in red) in *Mb*$^{+/+}$ versus *Mb*$^{-/-}$. (**L, O**) DNA/nuclei were co-stained

*Figure 4 continued on next page*

*Figure 4 continued*

blue with DAPI. (**M**) and (**P**) are brightfield images of (**K**) and (**N**), respectively. (**Q**) Quantification of CitH3 staining. Scale bars: 50 µm. Blue dashed lines refer to mean fluorescence intensities for uninjured dorsal skinfolds. All quantitative data are reported as means ± SD. n = 3 mice. *<0.05. Statistical significance in (**E**), (**J**), and (**Q**) was computed by unpaired Student's *t* test.

The online version of this article includes the following figure supplement(s) for figure 4:

**Figure supplement 1.** Nitroxidative stress in *Mb*$^{+/+}$ versus *Mb*$^{-/-}$ tissues 3 d after muscle pressure injury (mPI).

**Figure supplement 2.** Extracellular myoglobin increases reactive oxygen species (ROS) in and decreases the function of macrophages in vitro.

## Effects of iron chelation therapy on immune infiltration and regeneration

At 7 d post-injury, DFO-treated tissues displayed 156% greater abundance of immune infiltrate, according to brightfield and DAPI staining (p=0.0273; *Figure 7C–D and L*). Co-staining with iNOS showed strong co-localization with the infiltrate. iNOS is a marker of pro-inflammatory stimulation in many cell types, and it showed a 43% increase after DFO treatment (p=0.0183; *Figure 7A–D and K*). CD38, a marker of CD4+, CD8+, B, and natural killer cells, was not significantly different between saline- and DFO-treated mPI (p=0.0941; *Figure 6—figure supplement 3*). On the other hand, Mer tyrosine kinase (MerTK) exhibited a 170% increase in expression in DFO-treated tissues (p=0.0189; *Figure 7—figure supplement 1*). MerTK is a marker of monocytes and other specialized cell types. It has been shown to promote macrophage survival (*Anwar et al., 2009*) and phagocytic function (specifically efferocytosis) (*Scott et al., 2001*).

Later at day 10, immune infiltrate was still elevated in DFO-treated wounds (*Figure 7M*, p=0.0014 for histopathology and p=0.0004 for count of DAPI nuclei). Arginase-1 (Arg-1) is a marker of regenerative activity and promotes polyamine biosynthesis for wound healing. Arg-1 levels were twofold higher in the wound bed at day 10 (p=0.0269; *Figure 7N*) and much of the day 10 infiltrate was positive for Arg-1 (*Figure 7E–J and M–O*). Arg1-positive cells showed 119% greater distance of infiltration into injured tissue (p=0.0061; *Figure 7P*) compared to saline-treated control. Granulation was also dramatically improved. Granulation is the regenerative stage of wound healing, characterized by angiogenesis (formation of new capillaries) and proliferation of epithelial, endothelial, and fibroblast cells, along with continued presence of immune infiltrate. Angiogenesis is widely reported to increase after DFO treatment (*Duscher et al., 2015*; *Holden and Nair, 2019*), and DFO-treated mPI had a similar trend, evidenced by a twofold increase in small blood vessels (p=0.0038; *Figure 7—figure supplement 2A–G*).

## Quality and quantity of regeneration

DFO treatment improved the extent of muscle regeneration (*Figure 8A–F* and *Figure 7—figure supplement 2H*). Much of this improvement was complete by day 40 when treatment improved regeneration at the wound center (p=0.0345; *Figure 8E*) and wound edge (p=0.0121; *Figure 8F* and *Figure 8—figure supplement 1*). In saline-treated wounds, myoblastic cells were observed at 40 d (*Figure 8A and C*), indicating that muscle regeneration was still underway.

When we extended the study to 90 d, no further increase in panniculus regeneration was detected (from 40 to 90 d) in saline-treated mPI. Compared to healthy regeneration (*Figure 9A–C*, CTX-injured), the newly regenerated myofibers in saline-treated wounds had pathological morphology, seen from branched or split fibers, thin fibers, wavy instead of straight axes, and disjoint angles between different bundles of fibers (*Figure 9D–F*). Newly regenerated myofibers were distinguishable from preexisting or never injured myofibers by their expression of fluorescent proteins because our mice expressed the confetti transgene in Pax7-positive satellite cells and their progeny (*Nasir et al., 2022*; *Tierney et al., 2018*). DFO-treated tissues displayed a far lower frequency (p=0.0021) of morphological malformations than saline-treated tissues (*Figure 9G–J*). At the day 90 endpoint, DFO-treated wounds had significantly smaller gaps in the original region of muscle (46% smaller area of unregenerated muscle, p=0.0087; *Figure 9K*) compared to saline-treated.

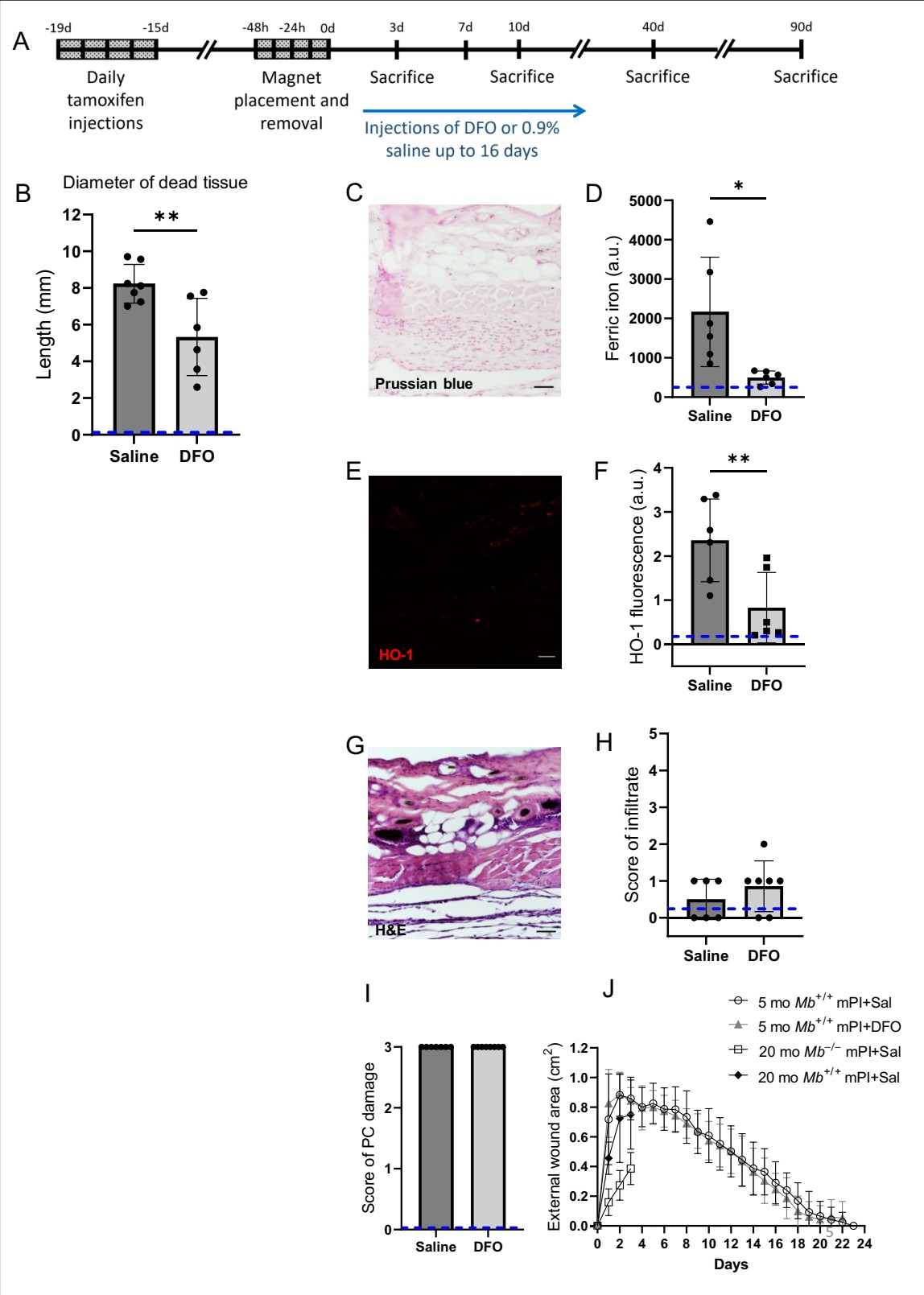

**Figure 5.** Deferoxamine (DFO) iron chelation therapy decreased iron deposits at day 3 after muscle pressure injury (mPI). (**A**) Experimental schedule shows confetti tamoxifen induction of fluorescence, mPI induction, treatment with DFO or saline, and tissue harvest. (**B**) Diameter of the dead region of the panniculus carnosus (PC) muscle in tissue sections from DFO- versus saline-treated mPI, 3 d post-injury. (**C**) Perls ' Prussian blue iron staining of DFO-treated wounds at the wound center. (**D**) Quantification of Perls ' staining, showing comparison against saline-treated mPI from *Figure 2I*.

*Figure 5 continued on next page*

*Figure 5 continued*

(**E**) Immunostaining of HO-1 in DFO-treated mPI. (**F**) Quantification of HO-1 staining, showing comparison against saline-treated mPI from *Figure 2L*. (**G**) H&E-stained sections of DFO-treated mPI at the wound center. (**H**) Histopathology scoring of immune infiltrate at all layers of the wound center at day 3, comparing DFO-treated versus saline-treated mPI, which appear in *Figure 2B and (D)*. (**I**) Confirmation that injuries were properly created, according to death of PC tissue at the center of the wound (histopathology scoring where 3 indicates pervasive tissue death). (**J**) Skin wound area following mPI in 5-month-old $Mb^{+/+}$ saline and DFO-treated mice, and in 20-month-old $Mb^{+/+}$ and $Mb^{-/-}$ saline-treated mice. Scale bars: 50 µm. Blue dashed lines refer to histology scores and mean fluorescence intensities for uninjured dorsal skinfolds. All quantitative data are reported as means ± SD. n = 6–7 mice. *<0.05, **<0.01. Statistical significance in (**B**), (**D**), (**F**), and (**H–J**) was computed by paired Student's *t* test.

The online version of this article includes the following figure supplement(s) for figure 5:

**Figure supplement 1.** Saline-treated muscle pressure injury (mPI) show increased TUNEL staining compared to deferoxamine (DFO)-treated mPI and cardiotoxin (CTX) injury.

**Figure supplement 2.** Deferoxamine (DFO) treatment did not significantly affect external wound sizes nor times to wound closure.

## Discussion

Our study established a mouse model of mPI that exhibited poor healing in the absence of any infection, hyperglycemia, or old age. With blood vessels collapsed or disrupted, the mPI had dead tissue persisting in the wound bed for more than a week, and regeneration was impaired. The debris field also contained high accumulations of iron, which motivated us to study myoglobin-knockout mice and iron chelation therapy. These experiments showed that the iron was myoglobin-dependent, and the myoglobin iron caused oxidative damage. The damage was consistent with lipid peroxidation, ETs, and ferroptosis. Finally, we showed that administering DFO after pressure caused decreased oxidative stress, decreased death of tissue margins, and improved muscle regeneration. The numbers of intact immune cells in mPI were positively correlated with regeneration and negatively correlated with oxidative damage, suggesting that excessive inflammation was not the primary pathology causing poor healing in these wounds.

Induction of mPI used a standard magnet method in mice, but instead of studying longitudinal re-epithelialization, we used tissue section timepoints to study the thin muscle layer of panniculus carnosus, also called the cutaneous muscle or the panniculus layer. This muscle regenerated slowly and incompletely from the magnet injury (*Figure 1H*), even under ideal conditions of youth, health, and lack of microbial pathogens. A hole remained in the panniculus layer at 40 d (in seven out of seven mice) and 90 d (in five out of five mice). For example, in *Figure 1D*, note that the adipose layer is immediately adjacent to loose areolar tissue at the wound center, indicating that the panniculus layer is absent. Thus, we obtained a normoglycemic noninfected animal model of impaired healing.

Mb knockout tested the hypothesis that myoglobin iron contributes to the pathologies of mPI. This hypothesis is based on analogy to poor-healing wounds that have hemolysis and poor drainage of hemoglobin (e.g., venous stasis), and also inspired by papers showing myoglobin is detectable in distal fluids after pressure ulcers (*Traa et al., 2019*; *Levine, 1993*; *Traa, 2019*). As expected, the iron deposits seen after wildtype mPI injury were absent from $Mb^{-/-}$. Similarly, $Mb^{-/-}$ had far less oxidative damage. Because Mb normally serves to supply oxygen in muscle tissue, one might expect $Mb^{-/-}$ mice to experience increased tissue death due to the hypoxia of ischemia, but prior work showed $Mb^{-/-}$ mice are 'surprisingly well adapted' to hypoxic conditions (*Schlieper et al., 2004*), capable of withstanding adrenergic stimulation (*Meeson et al., 2001*), exercise, and oxygen flux (*Gödecke et al., 1999*). In the mPI context, carrying wildtype Mb caused roughly twice as much tissue death as $Mb^{-/-}$. Some of that difference might be explained by developmental compensation (*Gödecke et al., 1999*) (e.g., 17% increase in capillary density), but we interpret that a large fraction of tissue death from mPI was downstream of Mb. Multiple other DAMPs including hemoglobin might have contributed to the iron accumulation and oxidative damage seen in the $Mb^{+/+}$ case, but removing Mb was sufficient to alleviate the overload.

The region of myoglobin-related damage extended beyond muscle. Given the potential toxicity of myoglobin during ischemia-reperfusion (*Garry et al., 1998*; *Gödecke et al., 1999*; *Meeson et al., 2001*; *Hazarika et al., 2008*; *Meisner et al., 2013*), we were not surprised that Mb knockout caused greater survival of muscle. However, we were surprised that most other layers of the wound improved too. Some types of ROS created by iron are highly reactive (e.g., hydroxyl radical from the Fenton reaction), so they act locally where they are created. Long-distance damage might arise from transport

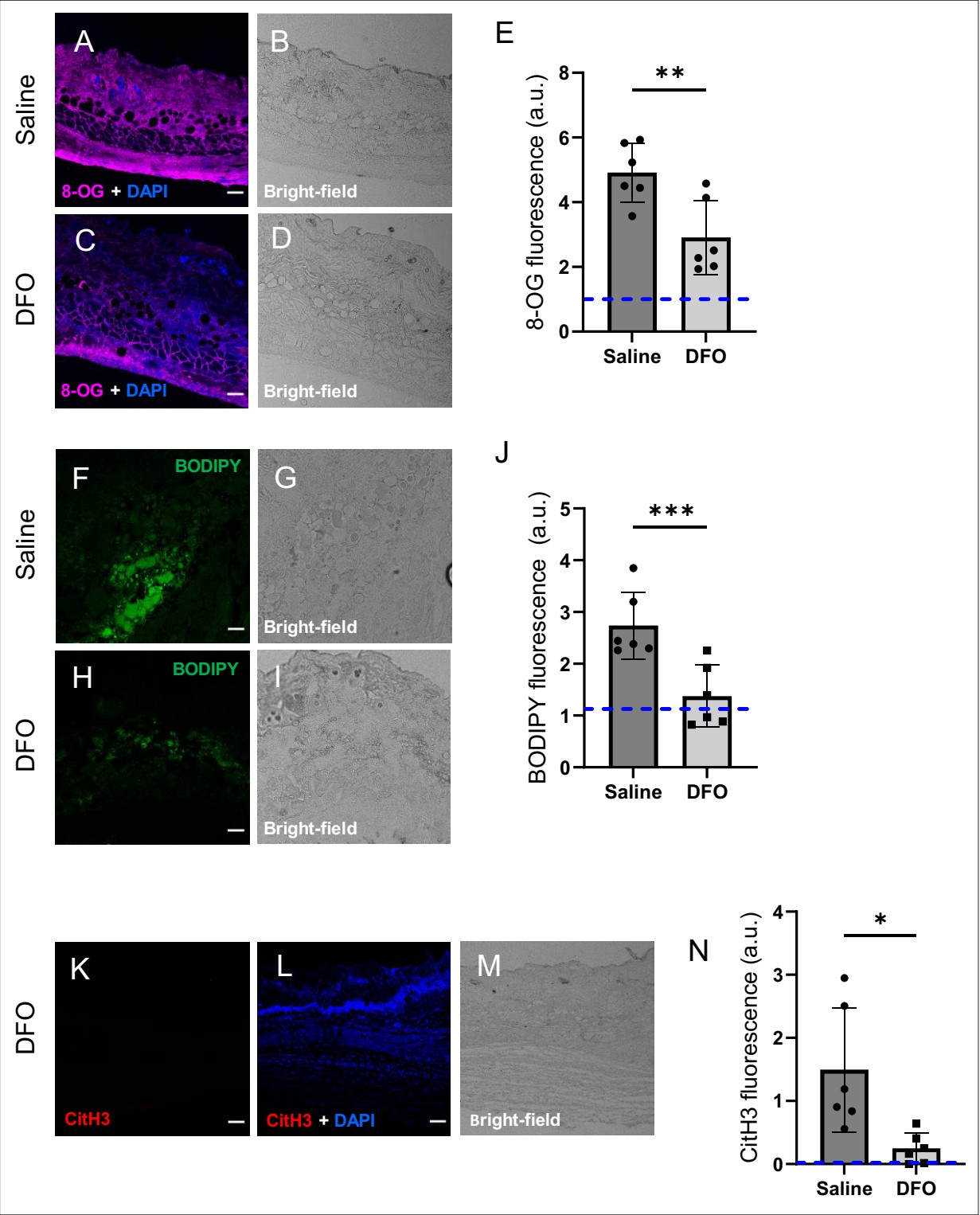

**Figure 6.** Deferoxamine (DFO) treatment improved the muscle pressure injury (mPI) microenvironment at early time point (day 3). (A–D) Immunostaining of 8-OG (for DNA damage) in saline-treated versus DFO-treated mPI. Nuclei were stained blue with DAPI. (B) and (D) are brightfield images of (A) and (C), respectively. (E) Quantification of 8-OG. (F–I) BODIPY staining (for lipid peroxidation) and brightfield in saline- versus DFO-treated mPI. (J) Quantification of BODIPY. (K) CitH3 immunostaining (L) with DNA/nuclear co-stain and (M) brightfield in DFO-treated mPI at day 3. (N) Quantification of CitH3 staining in DFO-treated versus the saline-treated mPI, which was analyzed in *Figure 2F and G*. Scale bars: 50 μm. Blue dashed lines refer to mean fluorescence intensities for uninjured dorsal skinfolds. All quantitative data are reported as means ± SD. n = 6 mice. *<0.05, **<0.01, ***<0.001. Statistical significance in (E), (J), and (N) was computed by paired Student's *t* test.

*Figure 6 continued on next page*

*Figure 6 continued*

The online version of this article includes the following figure supplement(s) for figure 6:

**Figure supplement 1.** Nitroxidative stress in deferoxamine (DFO) versus saline-treated tissues 3 d after muscle pressure injury (mPI).

**Figure supplement 2.** Injury response at day 3 after pressure.

**Figure supplement 3.** Comparing immune markers in cardiotoxin (CTX)-injured tissue, saline-treated muscle pressure injury (mPI), and deferoxamine (DFO)-treated mPI.

of the globin iron, or from milder ROS species, or both. Given the large effect sizes seen in the $Mb^{-/-}$ and DFO experiments, and given the small amount of muscle in mPI, we suspect that unappreciated mechanisms of amplification or vulnerability might be downstream of myoglobin iron in the mPI context.

Clearance of myoglobin iron can be seen in CTX injuries, which had destruction of muscle but no accumulation of iron, according to Perls' stain (*Figure 2J*). CTX and naniproin are snake venoms that cause myolysis (cytolysis) of myofibers (*Guardiola et al., 2017*; *Harvey et al., 1982*), and have been reported to spare nerve and vessel viability at the concentrations used (*Averin and Utkin, 2021*; *Wang et al., 2022*). One presumes that myolysis would cause myoglobin release, but continued function of the circulatory system may have allowed the myoglobin iron to be transported, diluted, and/or detoxified below the detection limit of Perls' stain. In contrast, levels of iron in mPI were greater than the detection limit for Perls' stain (*Figure 2I*), and many dysfunctions ensued. Pressure has been reported to cause disruption to arterial supply, venous clearance, and lymphatic drainage (*Gray et al., 2016*; *Kimura et al., 2020*; *Karahan et al., 2018*), although not always (*Stekelenburg et al., 2006*). Lymph vessels are occluded at very low pressure, and impaired lymphatic flow can persist after pressure is removed (*Worsley et al., 2020*). Impaired circulation and/or impaired drainage could cause waste factors such as myoglobin or iron to accumulate in the mPI, and could impair the influx of iron detoxification factors. Poor drainage also occurs in chronic ulcers of venous insufficiency, lymphedema, and other vascular diseases. Toxicity from globin iron might be a shared pathology among these wounds. mPI research might also benefit from analogy to known globin overload syndromes such as compartment syndrome, rhabdomyolysis, and hemoglobinopathies.

Pressure injury prevention usually refers to pre-injury prevention, which is to intervene with cushioning or turning before prolonged pressure occurs (*Sundin et al., 2000*). What we studied is post-pressure prevention because we asked whether tissue could be saved from dying by interventions beginning 12 hr after pressure had finished. In healthy young mice, magnets were placed and removed in 12 hr cycles as part of a 2-d process of creating a pressure ulcer. (See the detailed timeline in *Figure 5A*.) Drug injections began on the third day after pressure began, which we refer to as day 1 post-injury. The remarkable observation (*Figure 5B*) is that the amount of death on day 3 post-injury was not constant across the tissue sections, even though the induction of the pressure ulcers was identical. mPI that received post-pressure DFO showed significantly smaller diameters of muscle death than saline-treated comparisons. Our quantification used cross-sectional tissue slides, but a 35% decreased diameter of death equates to a 58% smaller circle area of death. We conclude that there is window of opportunity to intervene and save viability of tissue, after the established mechanisms of injury have occurred (such as mechanical, hypoxic, reoxygenation, and nutrient stress), but before the full extent of secondary progression and DAMP-induced stress have propagated. In this mouse model, the opportunity occurred between 12 hr and 72 hr after off-loading.

Secondary progression of wounds is well documented in thermal burns, where post-burn treatment can lessen the progression of partial-thickness burns toward full-thickness. However, secondary progression has not, to the best of our knowledge, been previously targeted for medical intervention in pressure ulcers (*European Pressure Ulcer Advisory Panel et al., 2019*). Mechanisms of secondary progression may include oxidative stress and ferroptosis (*Li et al., 2020*; *Li et al., 2021*), and we cannot rule out reperfusion injury, platelet activation, ETs, and DAMP-induced apoptosis/necroptosis (*Jagannathan and Tucker-Kellogg, 2016*).

CitH3, a marker of ETs, was elevated in the high-iron conditions (young and elderly mPI) and low in the three conditions that lacked concentrated deposits of iron (CTX, knockout, DFO-treated). This is consistent with the ability of heme, oxidative stress, or heme-activated platelets to trigger ETs (*Okubo et al., 2018*; *Ohbuchi et al., 2017*). ETs can aid in antimicrobial defense, but they are detrimental to

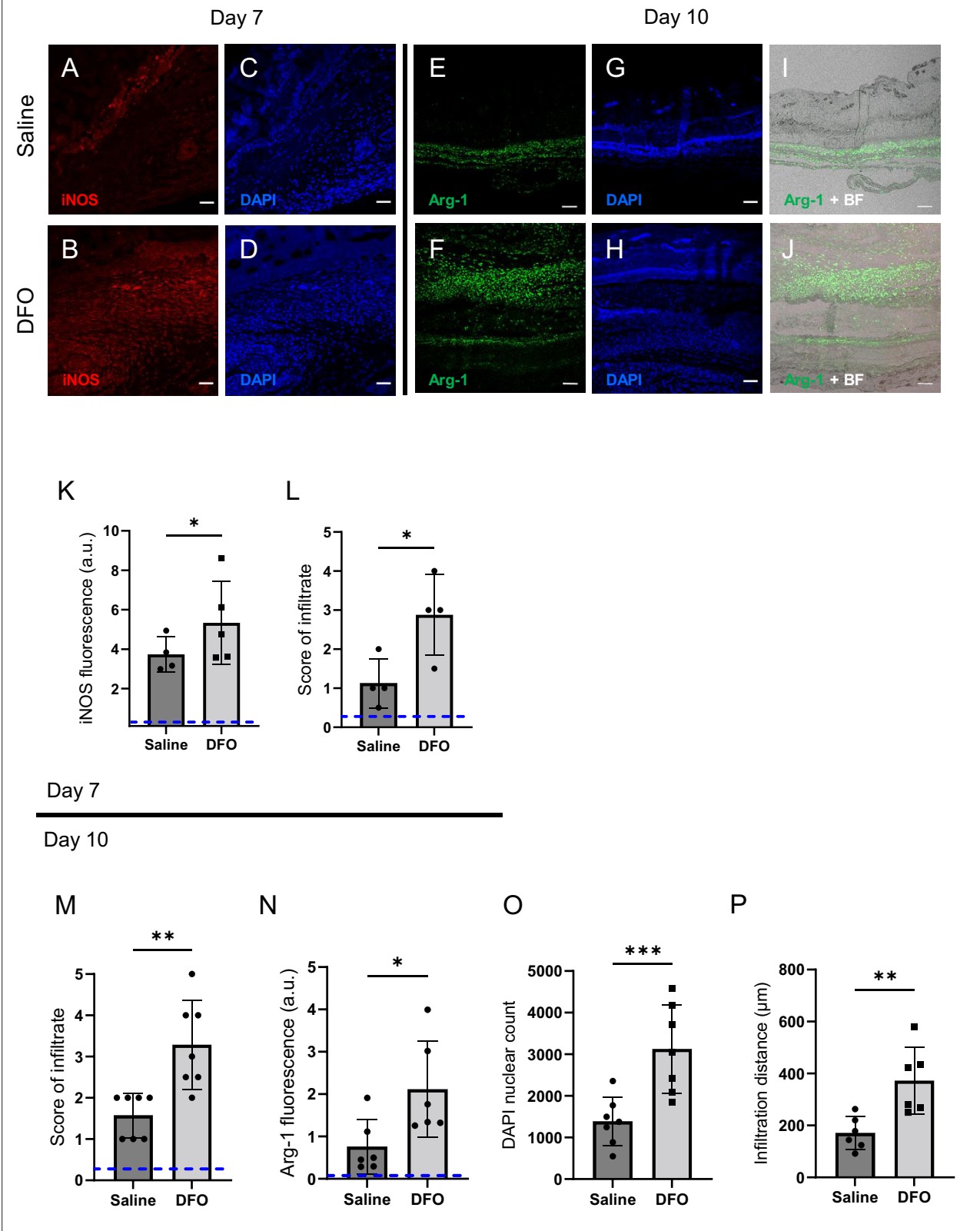

**Figure 7.** Deferoxamine (DFO) treatment improved immune infiltration (7 and 10 d after muscle pressure injury [mPI]). (**A, B**) Immunostaining of inducible nitric oxide synthase (iNOS, associated with pro-inflammatory activation) in saline- versus DFO-treated mPI at day 7 post-injury. (**C, D**) Nuclei are stained blue with DAPI to show the count of infiltrating cells. (**E, F**) Immunostaining of Arginase-1 (Arg-1) in saline- versus DFO-treated mPI at day 10 post-injury. (**G, H**) Nuclei are co-stained blue with DAPI. (**I, J**) Merged brightfield and Arg-1 immunostaining. (**K**) Quantification of iNOS staining at day 7

*Figure 7 continued on next page*

*Figure 7 continued*

between saline- and DFO-treated. (**L**) Scoring of immune infiltrate into the injured tissue at day 7. (**M**) Scoring of immune infiltrate into the injured tissue at day 10. (**N**) Quantification of Arg-1 staining. (**O**) Count of DAPI nuclei. (**P**) Distance of tissue infiltrated by Arg-1+ cells in saline- versus DFO-treated tissues. Scale bars: 50 μm. Blue dashed lines refer to histology scores and mean fluorescence intensities for uninjured dorsal skinfolds. All quantitative data are reported as means ± SD. n = 4–7 mice. *<0.05, **<0.01, ***<0.001. Statistical significance in (**K–P**) was computed by paired Student's *t* test.

The online version of this article includes the following figure supplement(s) for figure 7:

**Figure supplement 1.** Immune infiltration and function in deferoxamine (DFO) versus saline-treated tissues 7 d after muscle pressure injury (mPI).

**Figure supplement 2.** Deferoxamine (DFO) treatment improved angiogenesis and granulation following muscle pressure injury (mPI).

regeneration and contribute to many disorders of sterile inflammation. Our induction of CitH3 was likely sterile as well: the mouse environment was negative for 39 categories of pathogen (**Supplementary file 1h**, including common dermal microbiota *Pseudomonas aeruginosa* and *Staphylococcus aureus*), and no bacteria could be detected in or near the wound by Gram staining.

The persistence of dead tissue was particularly striking in our noninfected mPI wound. Viable immune cells were essentially absent from the pressure-injured muscles of young healthy mice at day 3. The wounds exhibited a normal spike in immune cell infiltration outside the wound margins, moving toward the wound (**Figure 2—figure supplement 1**), but intact infiltrate was not seen inside the compressed region, almost as if infiltration had been halted at the boundary. Initially we questioned whether the compressed architecture of the tissue might have occluded chemotaxis, but closer inspection revealed dead fragments and markers such as CitH3 in the compressed region. Our interpretation is that immune cells (tissue-resident and/or infiltrated) may have been present but then died. We do not know whether this death was necrotic or programmed. Possible explanations for the persistence of dead tissue, in addition to poor cell survival, are decreased migration and decreased efferocytosis/phagocytosis. Diseases characterized by the presence of plaque (atherosclerosis) and ectopic tissue (endometriosis) have impaired function of phagocytes (**Liu et al., 2019**; **Schrijvers et al., 2005**). Other syndromes with iron overload have poor function, migration, and/or survival of phagocytes (**Kao et al., 2016**; **Van Asbeck et al., 1984**; **Porto and De Sousa, 2007**). For example, in patients with thalassemia major, neutrophils and macrophages exhibit poor chemotaxis, defective lysis, and impaired phagocytosis (**Ballart et al., 1986**; **Martins et al., 2016**; **Cantinieaux et al., 1987**).

Intact immune cells were present at days 7 and 10 of mPI. DFO increased their abundance, especially in the loose areolar tissue, but also in muscle, fat, and skin (**Figure 7L and M**). One theory of nonhealing wounds is that excessive inflammation causes oxidative damage and blocks progress of granulation. Our mPI observations are not well aligned with that theory because DFO caused an increase in the immune infiltrate and iNOS staining, but it *decreased* the oxidative damage and the size of the hole. Another theory is that nonhealing wounds lack pro-regenerative M2 macrophages. Nothing observed about mPI in this first study would contradict that theory. Our work is also consistent with prior work that found iron-scavenging macrophages from an iron-loaded tissue displayed both inflammatory (iNOS, IFNγ) and alternately-activated (Arg1, IL10) markers (**Ali et al., 2020**). Prior studies also found that environmental iron can cause M2 macrophages to decrease (**Handa et al., 2019**) or increase (**Agoro et al., 2018**), depending on dose/context. Future studies can use our mPI timeline to design an immunobiology study that isolates mPI immune cells for detailed analysis.

Surface visibility of dead tissue was delayed until competent inflammation and granulation pushed it upward. This delay might give a false impression that slough was created by infection or inflammation. The slough detached spontaneously by day 15, but as it was pushed up, the slough remained attached to the healthy margins and bent the margins upward (**Figure 1E** and **Figure 6—figure supplement 2**). Since healthy margins organize the geometry of regeneration, distorting the margins might impair regeneration. Nonhealing wounds are sometimes believed to have excess proteolysis, but mPI raises the question of whether increased proteolytic activity might help prevent the extrusion process from distorting the healthy margins.

The toxicity of slough is a topic of clinical debate. In our mPI, re-epithelialization proceeded beneath the necro-slough and proliferative granulation tissue was adjacent to its center, so we conclude it was not cytotoxic. This is consistent with clinical practice to avoid debridement of an intact eschar unless there are signs of infection (**Manna et al., 2021**). Interestingly, our slough arose from a mass that was probably toxic when it formed. Toxicity at early timepoints is inferred from the oxidative damage at day 3 and indirectly from the phenomenon of secondary progression. It is possible that toxicity could

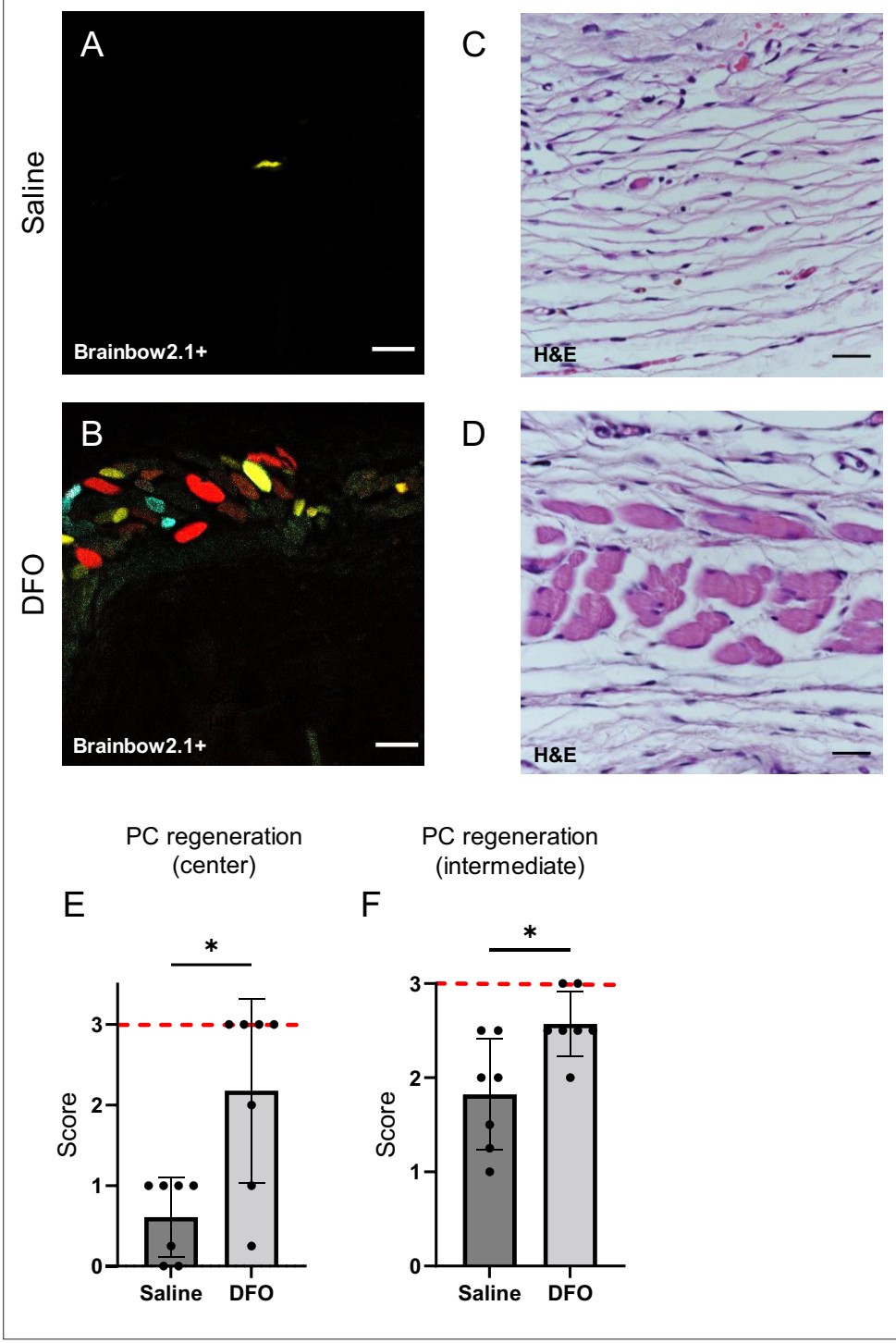

**Figure 8.** Deferoxamine (DFO) increased the extent of muscle regeneration at day 40. (**A, B**) Confocal fluorescent and (**C, D**) H&E-stained cross-sections of regenerated muscle fibers from saline- and DFO-treated muscle pressure injury (mPI), 40 d post-injury. Scale bars: 100 μm. (**E, F**) Histological scoring of panniculus carnosus (PC) muscle regeneration in saline- and DFO-treated mPI versus acute cardiotoxin (CTX)-injured tissues at the center and edge of the wound, 40 d post-injury. Red dashed lines refer to PC regeneration scores at center and intermediate regions after CTX injury at day 40. All quantitative data are reported as means ± SD. n = 7 mice. *<0.05, **<0.01. Statistical significance in (**E, F**) was computed by paired Student's *t* test.

The online version of this article includes the following figure supplement(s) for figure 8:

**Figure supplement 1.** Failure of muscle regeneration after muscle pressure injury (mPI) is ameliorated by deferoxamine (DFO) treatment.

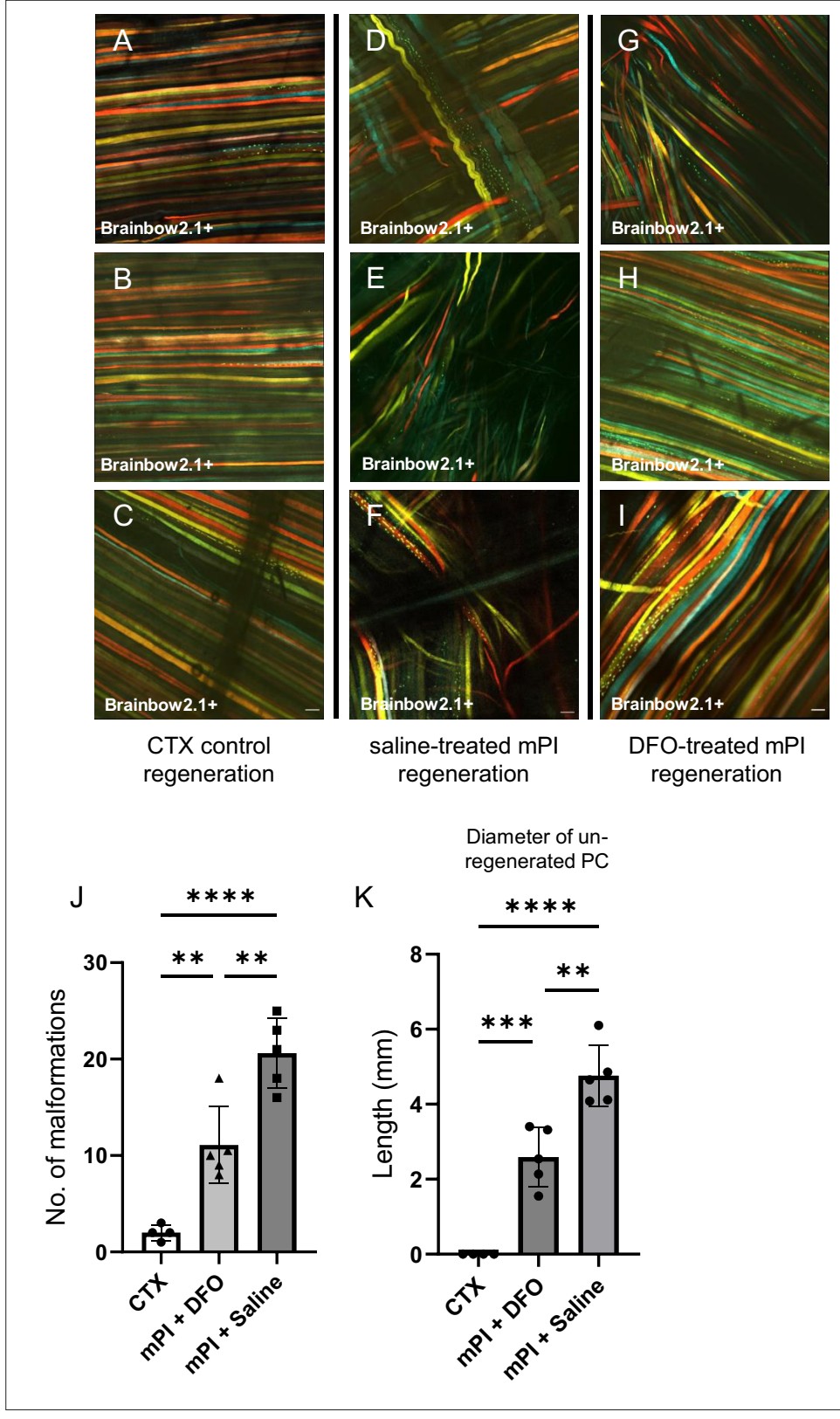

CTX control regeneration

saline-treated mPI regeneration

DFO-treated mPI regeneration

**Figure 9.** Deferoxamine (DFO) improved muscle morphology. Confocal microscopy of confetti-labeled muscle regeneration in ex vivo tissue blocks. Note that unlabeled tissue types such as blood vessels and hair create black shadows on top of the images. (**A–C**) Left column shows healthy regeneration in the control condition, toxin-induced injury, at 40 d. Experiment was halted at 40 d for cardiotoxin (CTX) only due to complete regeneration.

*Figure 9 continued on next page*

*Figure 9 continued*

(**D–F**) Middle column shows regenerated muscle fibers of saline-treated muscle pressure injury (mPI), 90 d post-injury. Note the presence of non-parallel fibers, bent fibers, and split fibers (i.e., fibers with one or more branches). (**G–I**) Right column shows regenerated muscle fibers of DFO-treated mPI, 90 d post-injury. Scale bars: 100 μm. (**J**) Quantification of muscle fiber malformations. (**K**) Diameter of gap (unregenerated region) in panniculus carnosus (PC) muscle layer at day 90 post-mPI between saline- and DFO-treated wounds. All quantitative data are reported as means ± SD. n = 4–5 mice. \*\*<0.01, \*\*\*<0.001, \*\*\*\*<0.0001. Statistical significance in (**J, K**) was computed by one-way ANOVA with Tukey's post hoc test.

disappear spontaneously if ROS are driven by an energy source that gets depleted (e.g., hydrogen peroxide, ATP, mitochondrial membrane potentials).

Muscle regeneration improved with DFO treatment, evidenced by smaller holes in the muscle layer at multiple timepoints, and improved muscle morphology at the endpoint. The limited muscle regeneration that did occur in mPI without DFO displayed frequent myofiber defects such as branching and nonparallel alignment. Split fibers are considered defective regeneration (*Eriksson et al., 2006*) and are especially vulnerable to reinjury (*Pichavant et al., 2016*). Loss of the panniculus muscle layer causes the skin and adipose to experience harsher stress and strain (*Nasir et al., 2022*; *Soh et al., 2020*), even though the panniculus is a thin component of the tissue. Previous studies of muscle pressure injury considered larger muscles (*Ahmed et al., 2016*; *Siu et al., 2009*; *Wassermann et al., 2009*; *Salcido et al., 1994*; *Salcido et al., 2007*), and the impact of DFO should be retesting in other geometries. The panniculus layer is highly relevant to human pressure injuries because humans have a panniculus layer at the heel (*Nasir et al., 2022*; *Cichowitz et al., 2009*), and the heel is one of the most common locations for high-stage pressure injuries.

Skin regeneration was not significantly affected by subcutaneous DFO (*Figure 5J* and *Figure 5— figure supplement 2*). Previous work found DFO administered via a transdermal patch accelerated skin closure (*Duscher et al., 2015*), but these studies used topical delivery, as well as using a different mouse background. Other work studied the antioxidant and angiogenic benefits of DFO in preclinical models of ischemia-reperfusion injury (*González-Montero et al., 2018*; *Morris et al., 1993*; *Yang et al., 2020*; *Peeters-Scholte et al., 2003*) and related conditions (*Qayumi et al., 1994*). We observe similar effects in muscle.

Caveats to this work include the following. Mouse wounds differ from human in several obvious ways, such as having denser hair follicle coverage, faster kinetics of DFO, different timing and ratio of phagocytic cells (*Zomer and Trentin, 2018*), and higher propensity for wound contraction. Although mice can heal skin wounds by contraction, our multilayer mPI exhibited minimal contraction, as seen from prior work (*Nasir et al., 2022*) and from the hole in the panniculus muscle layer at day 90. To decrease the impact of hair follicles on results, we placed magnets on regions of skin that were in the resting phase of the hair cycle. Another limitation of our study is the limited number of myoglobin knockout mice (n = 3) compared with n = 7 for key timepoints of $Mb^{+/+}$ analysis. This was necessary because $Mb$ germline knockout had high rates of embryonic lethality. Despite the smaller sample size, several readouts were statistically significant.

## Conclusion

Myoglobin iron contributed to the size, severity, oxidative damage, and poor healing of mPI. DFO injections prevented death of tissue at the margins of the wound, when administered starting 12 hr after pressure was removed. Intervention to block secondary progression is a dramatic new opportunity for pressure injuries, like burns or strokes, to be stabilized shortly after injury, to decrease further loss of salvageable tissue. Iron chelation therapy has side effects when used systemically, so muscle wounds might be safer than vascular wounds for future research on human biodistribution (in conjunction with dressings and polymer scaffolds). Also, DFO led to higher quality of muscle, with straighter and more parallel muscle fibers, which should improve muscle viability. Healthy muscle is a crucial defense against wound recurrence, when a patient eventually puts weight on the same location again.

## Methods

### Mice

Animal experiments were approved by the institutional animal care and use committee (IACUC SHS/2016/1257) of SingHealth, Singapore. To conditionally label Pax7+ muscle satellite stem cells in C57BL6 mice, the Pax7-Cre-ER$^{T2}$ mouse (JAX stock #012476, Jackson Laboratory, ME) was crossed with the Brainbow2.1 (confetti; JAX stock #017492, Jackson Laboratory) mouse. Pax7-Cre-ERT2 provides the Cre-ERT2 transgene downstream of the Pax7 stop codon, thereby limiting the Cre-ERT2 expression to Pax7+ cells. The Cre-ER$^{T2}$ system provides tamoxifen-dependent induction of Cre recombinase, so that affected cells carry heritable rather than transient modification. Upon tamoxifen treatment, the Cre induction causes recombination of the confetti construct at its loxP loci, leading to gene expression of one of the four fluorescent proteins in the construct. The fluorescent proteins are mCerulean (CFP$^{mem}$), hrGFP II (GFP$^{nuc}$), mYFP (YFP$^{cyt}$), and tdimer2 (*Mendonça et al., 2016*) (RFP$^{cyt}$). CFP$^{mem}$ contains a localization sequence enabling its transport to the myofiber membrane (sarcolemma) while GFP$^{nuc}$ contains a nuclear localization sequence. YFP$^{cyt}$ and RFP$^{cyt}$ have no localization sequences and are expected to localize to the cytoplasm. Experimental mice include 5-month-old adult mice and 20-month-old elderly mice.

Myoglobin knockout mice (homozygous in the germline) were created by Cyagen (CA) using CRISPR/Cas9 nuclease-mediated genome editing (*Qin et al., 2015*; *Qin et al., 2016*). 20-month-old mice (n = 3) were used in the knockout experiments. The myoglobin knockout mice were not injected with tamoxifen.

Mice were euthanized via $CO_2$ inhalation, followed by cervical dislocation, at timepoints of 3, 7, 10, 40, and 90 d following completion of the injury protocol (pressure or toxin). Pressure injury via magnetic compression created two wounds (right and left) on the dorsum of each mouse. Similarly, two CTX injections were performed on each mouse at the right and left dorsal skinfold. Tissue samples were isolated, and one wound tissue (from either pressure or toxin) was fixed in 4% paraformaldehyde at 4°C for 8 hr, and then embedded in paraffin. The remaining wound tissue was snap-frozen with isopentane in a liquid nitrogen bath.

### Murine pressure injury model

Mice were shaved and remaining hair was removed by hair removal cream (Veet, Reckitt, Slough, England) prior to injury. Muscle pressure ulcers were created in 4–5-month-old transgenic mice by applying a pair of ceramic magnets (Magnetic Source, Castle Rock, CO, part number: CD14C, grade 8) to the dorsal skinfold. The magnets were 5 mm thick and 12 mm in diameter, with an average weight of 2.7 g and pulling force of 640 g. One of the elderly mice lacked sufficient skinfold for 12 mm magnets, so in that mouse and in its age- and sex-matched control, we used 5 mm magnets (neodymium magnets with a 1 mm spacer on one side; Liftontech Supreme Pte Ltd, Singapore). Because hair follicle (HF) stem cells contribute to wound healing, we attempted to minimize the HF differences by synchronizing the hair growth cycle and applying injury only to skin with HFs in telogen phase (non-pigmented skin). In the case where pigmented skin could not be avoided during magnet placement, that is, one of the two wounds fell on skin with active hair cycle, that half was excluded from further analysis. Pressure ulcer induction was performed in two cycles. Each cycle was made up of a 12 hr period of magnet placement followed by a 12 hr period without magnets (*Figure 1— figure supplement 1*). This procedure induces two pressure wounds on the back of the mouse, on the left and right side of the dorsal skinfold. The dorsal skinfold includes skin, adipose tissue, panniculus carnosus muscle, and loose areolar tissue. The mice were given an analgesic (buprenorphine at 0.1 mg/kg; Bupredyne, Jurox Animal Health, New Zealand) prior to magnet placement and again prior to magnet removal. Timepoints post-injury are measured from the end of last magnet cycle. No dressings were used, and no debridements were performed.

To treat the pressure injuries, mice were subcutaneously injected with DFO while control mice were injected with 0.9% saline (n = 7 each for day 3, 10, and 40 timepoints, n = 4 for day 7, and n = 5 for day 90; *Supplementary file 1e*). The study was initially conducted using 3, 10, and 40 d. When the day 7 and 90 timepoints were added to the study, we already knew that DFO treatment would give a large effect size. The updated effect size caused our power calculation to give a smaller sample size for days 7 and 90. The day 7 timepoint was added later because of the dramatic changes observed between days 3 and 10, and the day 90 timepoint was added later because day 40 mPI showed signs

of ongoing regeneration (myoblastic cells and immature myofibers), indicating that steady state had not yet been reached at day 40.

DFO was administered twice per day at 30 mg/kg body weight (or 0.9% saline for control animals) for 16 d or until mouse sacrifice, whichever was sooner. Under anesthesia, the dorsal skinfold (cranial to the wound) was pulled away from the spine to make a 'tent,' and the needle was inserted into the tent (over the spine, near the left and right wounds). The first DFO treatment was given 12 hr after completion of the second and final cycle of pressure (*Figure 5A*). The dosing rationale was as follows: the recommended dose of DFO for iron overload in human patients is 20–60 mg/kg per day (maximum 80 mg/kg; *Barata et al., 1996*). Mice metabolize DFO faster than humans, so we took the 60 mg/kg per day human dosing and divided it into two half-doses per day.

## Murine cardiotoxin injury model

To induce an acute injury in the panniculus carnosus muscle in the skinfold, mice were shaved and fur removed by hair removal cream (Veet, Reckitt, Slough, England) prior to injury. The mice were anesthetized and 30 µl of toxin at 10 µM concentration was injected intramuscularly into the panniculus carnosus of the dorsal skinfold of each wound (left and right). Because commercial distribution of CTX from *Naja mossambica* was discontinued, some mice were injected with naniproin instead, while the term 'cardiotoxin' has been used for the entire cohort. Naniproin is a *Naja nigricollis* homolog of CTX with 95% sequence identity. Naniproin induces myolysis comparable to CTX. All toxin-injected mice were given an analgesic (buprenorphine at 0.1 mg/kg; Bupredyne, Jurox Animal Health) prior to injection. To control for the post-injury injections given to mPI mice, the toxin-injured mice were injected subcutaneously with 0.9% saline (n = 7) daily for 16 d or until mouse sacrifice, whichever was sooner (*Figure 1—figure supplement 1*). Tissues were harvested at day 3, 10, or 40 (n = 4). Uninjured healthy tissues were also collected as controls (n = 4).

## Pressure wound assessment

External wounds were assessed daily with a digital calliper (Tactix, CO, USA) positioned at the borders of the wounds to measure the length and width of each wound. In addition, each wound was photographed with an Olympus pocket camera alongside a wound mapping marker (KISS Healthcare Inc, CA). Measurements and photographs were obtained until re-epithelialization (external wound closure). Wound area was computed by ImageJ after manual segmentation of the wound margins.

## Histopathology scoring of H&E-stained sections

Paraformaldehyde-fixed paraffin-embedded tissue sections were stained with hematoxylin (Shandon Gill Hematoxylin 2; Thermo Fisher, MA) and eosin (Eosin Y; Sigma-Aldrich, MO) according to the manufacturer's instructions. H&E-stained slides were blinded and scored on the extent of tissue death, immune infiltration, granulation, and regeneration. The scoring for tissue death was defined as follows: 0, healthy tissue with zero or minimal death (<10% tissue area dead); 1, mild death (11–33%); 2, moderate death (34–67%); and 3, extensive tissue death (>67%). Death was identified by karyolysis, karyorrhexis, and acidification (eosinification). Scoring for tissue regeneration was defined as follows: 0 (minimal, i.e., <10% regenerated), 1 (mild, i.e., 11–33% regenerated), 2 (moderate, i.e., 34–67% regenerated) to 3 (extensive, i.e.,>67% regenerated). Scoring for granulation was defined as follows: 0, normal tissue with neither granulation nor neo-angiogenesis; 1, minimal granulation (less than 10% of tissue area); 2, mild granulation (10–25% of tissue area); 3, moderate granulation (26–50% of tissue area); 4, high granulation (51–75% of tissue area); and 5, extensive granulation (more than 75% of tissue area consisting of granulation tissue). Likewise, the scores for immune infiltration were defined as follows: 0, normal tissue without immune infiltration; 1, minimal immune infiltration (in less than 10% of tissue); 2, mild immune infiltration (in 10–25% of tissue); 3, moderate immune infiltration (in 26–50% of tissue); 4, high immune infiltration (51–75% of tissue area); and 5, extensive immune infiltration (more than 75% of tissue infiltrated by immune cells). Therefore, on the scales for tissue death, regeneration, granulation, and immune infiltration, an uninjured tissue would receive scores of 0, 3, 0, and 0, respectively. Scoring was performed for all treatments and all timepoints. The scoring of death, immune infiltration, and regeneration was performed for the overall wound and each tissue layer.

## Perl's Prussian blue staining

8 µm sections of fixed, paraffin-embedded tissues were deparaffinized and rehydrated prior to staining. The sections were then incubated for 5 min in Perl's Prussian blue iron stain (mix of potassium ferrocyanide and hydrochloric acid; Abcam, Cambridge, UK, ab150674) rinsed in water, and counter-stained in nuclear fast red (Abcam, Singapore, ab150674). Iron deposits were observed as deep blue dots or speckles, and the number and fractional area occupied by deep blue pixels were quantified by thresholding in MATLAB (MathWorks, MA).

## Confocal microscopy

Tissue samples were imaged using a Zeiss LSM710 confocal microscope (Carl Zeiss, Oberkochen, Germany) and Olympus FV3000 laser scanning confocal microscope (Olympus, Tokyo, Japan). Excitation and detection wavelengths used for the respective fluorophores were: $CFP^{mem}$: Ex. 457 nm and Em. 466–495 nm, $GFP^{nuc}$: Ex. 488 nm and Em. 498–510 nm, $YFP^{cyt}$: Ex. 515 nm and Em. 521–560 nm, $RFP^{cyt}$: Ex. 559 nm and Em. 590–650 nm. Images were processed and analyzed using Fiji (ImageJ) software. The number of muscle fiber malformations in each mouse was quantified by counting the malformations in a field of view, and averaging over four fields of view for each mouse.

## Immunofluorescence staining

10 µm fixed paraffin-embedded sections or cryosections were blocked with 10% normal serum and permeabilized with 0.2% Tween 20. Staining was performed using antibodies against 8-oxoguanine (8-OG or 8-hydroxy-2'-deoxyguanosine; Abcam, ab206461), F4/80 (Santa Cruz Biotechnology Inc, TX, sc-52664), Arginase-1 (Arg1; Abcam, ab60176, and Proteintech Group Inc, IL, 16001-1-AP), MerTK (Abcam, ab95925), inducible nitric oxide synthase (iNOS; Cell Signalling Technology, MA, #13120), citrullinated histone H3 (citH3; Abcam, ab5103), nitrotyrosine (Abcam, ab7048), and myoglobin (Cell Signaling Technology, 25919S). For detection, we used Alexa Fluor 488-, 594-, and 647-conjugated secondary antibodies (Abcam, ab150129, ab150064, and ab150075, respectively) raised in appropriate species for the experiments. The slides were mounted with Vectashield Hardset with DAPI (Vector Laboratories, CA) and images were acquired on a Leica TCS SP8 confocal microscope (Leica Microsystems, Wetzlar, Germany) and analyzed using LAS X software. For quantification, each wound consists of 512 × 512 image frames, covering all layers of the wound, and the number of frames depends on the size of the wound. Each 512 × 512 image frame was thresholded using Fiji (ImageJ) as outlined by *Shihan et al., 2021*, and the mean fluorescence intensity was computed. The mean intensity for each wound is computed by taking the mean of the frames. Text description has been added if specific layers were disproportionately responsible for the intensity.

## BODIPY 581/591 C11 detection

10 µm tissue cryosections were fixed in 4% paraformaldehyde, blocked with 10% normal serum, and permeabilized with 0.2% Tween 20. Tissue sections were stained with 10 µM BODIPY 581/591 C11 probe (Thermo Fisher, D3861). After 30 min in the dark, the slides were mounted with Vectashield Hardset with DAPI (Vector Laboratories) and viewed under a Leica TCS SP8 confocal microscope (Leica Microsystems). Each wound section was imaged from edge to edge with consecutive frames.

## TUNEL staining

Deparaffinized wound sections of CTX-injured or saline-treated mPI and DFO-treated mPI (8 µm sections) were stained with the DeadEnd Fluorometric TUNEL System (G3250; Promega, WI) according to the manufacturer's protocol. DAPI was used for nuclear staining. Images were visualized and captured with a Leica TCS SP8 confocal microscope (Leica Microsystems).

## Protein extraction and detection

Snap-frozen tissues were homogenized using MAGNAlyser beads (Roche Life Science, Penzberg, Germany) in RIPA lysis buffer (Sigma-Aldrich) with protease inhibitors and phosphatase inhibitors (Nacalai Tesque Inc, Kyoto, Japan). The samples were centrifuged for 20 min and the supernatant homogenate collected. Protein concentrations were determined using the Pierce BCA reagent (Thermo Fisher).

Luminex ELISA was carried out for the detection of murine interleukins, chemokines, and growth factors (R&D Systems, MN; LXSAMSM-21). Sample preparation followed the manufacturer's protocol, and analytes were read using the MAGPIX Platform (Luminex Corporation, TX).

For blotting, 30 µg samples were separated by SDS-PAGE (Bio-Rad Laboratories Inc, CA) and transferred to PVDF membranes (Bio-Rad Laboratories Inc). The PVDF membranes were blocked with 5% blotting grade blocker (Bio-Rad Laboratories Inc) at room temperature (RT) for 1 hr and then incubated with the primary antibodies (anti-myoglobin antibody, 25919S, Cell Signaling Technology) overnight at 4°C. Then, the membranes were washed with TBST at RT and incubated with the secondary antibody for 1 hr. Protein bands were detected with Pierce ECL substrate (Thermo Fisher) according to the manufacturer's instructions. Data were analyzed with the ChemiDoc Imaging System (Bio-Rad Laboratories Inc).

## Iron assay

An iron assay kit (Abcam; ab83366) was used to measure total iron (ferrous [$Fe^{2+}$] and ferric [$Fe^{3+}$]) in the uninjured tibialis anterior (calf) muscles of $Mb^{-/-}$ and $Mb^{+/+}$ mice. Snap-frozen muscle tissues were homogenized as previously described, and the assay was performed according to the manufacturer's instructions.

## RAW264.7 monocyte cell culture and differentiation

We cultured RAW264.7 macrophages (TIB-71, ATCC) in High Glucose DMEM with pyruvate (Gibco, UK) supplemented with 10% fetal bovine serum (FBS; GE Healthcare, USA), 100 units/ml penicillin, 100 µg/ml streptomycin, 0.25 µg/ml Amphotericin B (Fungizone, PSF; GE Healthcare), and 5 µg/ml Plasmocin (GE Healthcare) at 37°C under 5% $CO_2$. RAW264.7 monocytes were pre-incubated for 24 hr with 10 ng/ml recombinant murine IFNγ (Stem Cell Technologies, Vancouver, Canada) with 100 ng/ml lipopolysaccharide (LPS; Sigma-Aldrich) to resemble pro-inflammatory or M1 activation (*Cao et al., 2017*).

To dose the myoglobin treatment, we first used a mouse myoglobin ELISA kit (Abcam, ab157722) to obtain the myoglobin concentrations in the calf (67.4 µg/ml) and bicep (40.8 µg/ml) muscles of our mice. From this, we chose 15, 50, and 150 µg/ml to be the treatment concentrations of myoglobin (M0630 from horse muscle, Sigma-Aldrich) for the RAW264.7 cells (24 hr after pro-inflammatory treatment). Negative and positive controls were provided by untreated (0 µg/ml myoglobin) and menadione-treated macrophages. The cells were treated for 24 hr in the different conditions prior to performing assays for efferocytosis and oxidative stress.

## In vitro assays

After pro-inflammatory pretreatment and myoglobin (or control) treatment, the RAW264.7 cells were characterized as follows. The cells were measured for oxidative stress using the ROS-Glo $H_2O_2$ assay according to the manufacturer's instructions (G8820, Promega). Luminescence was measured using a Tecan Spark M10 microplate reader (Tecan, Männedorf, Switzerland). The cells were also measured using an efferocytosis assay kit (Cayman Chemical, MI) according to the manufacturer's instructions. Briefly, apoptotic bodies were created from staurosporine-treated C2C12 myoblasts (CRL-1772, ATCC) and labeled with CFSE fluorescence. Apoptotic bodies (or cell-free media control [CFM]) were added to wells for 16 hr prior to aspiration of the unengulfed apoptotic bodies, washing, and measurement. Absolute fluorescence was measured as integrated intensity for each well using a microplate reader (Tecan Spark M10, Männedorf, Switzerland) with excitation at 490 nm and emission at 525 nm.

RAW264.7 monocyte and C2C12 myoblast cell lines were authenticated using a commercial STR profiling service (Mouse Cell Authentication Service, ATCC 137-XV) and tested negative for mycoplasma using the MycoAlert PLUS mycoplasma detection kit (LT07-705; Lonza, Basel, Switzerland).

## Statistical analyses

Sample sizes for the mouse studies were determined using a power analysis for comparing two means (two-sample, two-sided equality).

In the analysis of DFO treatment in adult 5-month-old mice at multiple timepoints, treated mice were paired with age- and sex-matched controls. Therefore, significance was measured using a paired test (two-tailed Student's *t* test for mPI + DFO versus mPI + saline). Mice were not paired for the

other comparisons (i.e., adult 5-month-old mPI + saline versus CTX + saline cohorts2, and elderly 20-month-old $Mb^{-/-}$ mPI + saline versus $Mb^{+/+}$ mPI + saline cohorts), and statistical significance was analyzed using an unpaired two-tailed Student's $t$ test. For multiple comparisons in *Figure 9*, a one-way ANOVA was followed by the Tukey's post hoc test. For multiple comparisons in the Luminex assays (*Supplementary file 1df and g*), the Student's $t$ test was performed for each analyte, followed by the Bonferroni–Dunn correction for multiple hypothesis testing. Tests and plots were generated by GraphPad Prism (version 9.0.0 for Windows, GraphPad Software, CA). $*p<0.05$, $**p<0.01$, $***p<0.001$, and $****p<0.0001$. 'ns' means not significant.

## Acknowledgements

We acknowledge funding from the Singapore Ministry of Health's National Medical Research Council (NMRC/OFIRG/0007/2016) and also from the Singapore Ministry of Education's Tier2 Grant (MOE2019-T2-1-138) and the National Research Foundation, Prime Minister's Office, Singapore, under its Campus for Research Excellence and Technological Enterprise (CREATE) program, through Singapore MIT Alliance for Research and Technology (SMART): Critical Analytics for Manufacturing Personalised-Medicine (CAMP) Inter-Disciplinary Research Group.

We thank Manjunatha Kini and Koh Cho Yeow for naniproin; Ann-Marie Chacko, Tham Jing Yang, and Sarina Heng for detection methods; N Suhas Jagannathan for image analysis; Aadya S Deshpande, Korn Laongkul, Khadijah Zulkifli, and Colin Nicholas Sng for assistance; Peter TC So, Paul Matsudaira, Jasmine Chin, Mynn Varela, Ruklanthi de Alwis, Keshmarathy Sacadevan, the Mechanobiology Institute (Singapore) Microscopy facility, and the SingHealth Advanced BioImaging facility for advice.

## Additional information

### Funding

| Funder | Grant reference number | Author |
| --- | --- | --- |
| National Medical Research Council | NMRC/OFIRG/0007/2016 | Lisa Tucker-Kellogg |
| Ministry of Education - Singapore | MOE2019-T2-1-138 | Lisa Tucker-Kellogg |
| National Research Foundation Singapore | CAMP IRG | Hans Heemskerk Lisa Tucker-Kellogg |

The funders had no role in study design, data collection and interpretation, or the decision to submit the work for publication.

### Author contributions

Nurul Jannah Mohamed Nasir, Investigation, Data analysis, Writing; Hans Heemskerk, Investigation, Methodology; Julia Jenkins, Investigation, Project administration; Nur Hidayah Hamadee, Sample preparation; Ralph Bunte, Methodology; Lisa Tucker-Kellogg, Conceptualization, Supervision, Funding acquisition, Writing

### Author ORCIDs

Nurul Jannah Mohamed Nasir ⬤ http://orcid.org/0000-0003-3700-9036
Nur Hidayah Hamadee ⬤ http://orcid.org/0009-0002-8993-8419
Lisa Tucker-Kellogg ⬤ http://orcid.org/0000-0002-1301-7069

### Ethics

Animal experiments were approved by the institutional animal care and use committee (IACUC SHS/2016/1257) of SingHealth, Singapore.

### Decision letter and Author response

Decision letter https://doi.org/10.7554/eLife.85633.sa1
Author response https://doi.org/10.7554/eLife.85633.sa2

## Additional files

### Supplementary files
• Supplementary file 1. Supplementary tables. (**a**) Injuries to the panniculus carnosus muscle from CTX and mPI have comparable diameters at day 3, but are significantly different at day 10. (**b**) Schematic showing differences in injury response and regeneration between cardiotoxin (CTX, acute injury) and muscle pressure injury (mPI, chronic wound). (**c**) External wound area in *Myoglobin*$^{-/-}$ and age- and sex-matched *Myoglobin*$^{+/+}$ mice in the initial days following mPI, using 5 mm magnets. (**d**) Luminex measures of various cytokines, chemokines and growth factors between *Myoglobin*$^{+/+}$ and *Myoglobin*$^{-/-}$ tissues, three days after mPI. (**e**) Treatment arms for 5-month-old mice with mPI. (**f**) Luminex measures of various cytokines, chemokines and growth factors between saline- and DFO-treated, three days after mPI. (**g**) Luminex measures of various cytokines, chemokines and growth factors between saline- and DFO-treated, ten days after mPI. (**h**) Specific pathogen free status of animal housing facility.

• MDAR checklist

### Data availability
All data including primary images, quantification spreadsheets, and mouse tables have been uploaded to Zenodo at doi: https://doi.org/10.5281/zenodo.7754554.

The following dataset was generated:

| Author(s) | Year | Dataset title | Dataset URL | Database and Identifier |
|---|---|---|---|---|
| Nasir NJM, Heemskerk H, Jenkins J, Hamadee NH, Bunte R, Tucker-Kellogg L | 2023 | Myoglobin-derived iron causes wound enlargement and impaired regeneration in pressure injuries of muscle | https://doi.org/10.5281/zenodo.7754554 | Zenodo, 10.5281/zenodo.7754554 |

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

# Appendix 1

## Appendix 1—key resources table

| Reagent type (species) or resource | Designation | Source or reference | Identifiers | Additional information |
| --- | --- | --- | --- | --- |
| Strain, strain background (*Mus musculus*, C57BL6, male and female) | Pax7-Cre-ERT2 | Jackson Laboratory | #012476 | Transgenic mice |
| Strain, strain background (*M. musculus*, C57BL6, male and female) | Brainbow2.1 | Jackson Laboratory | #017492 | Transgenic mice |
| Strain, strain background (*M. musculus*, C57BL6, male and female) | Myoglobin-KO | Cyagen | | Germline knockout mice |
| Cell line (*M. musculus*) | RAW264.7 | ATCC | TIB71 | |
| Cell line (*M. musculus*) | C2C12 | ATCC | CRL-1772 | Myoblasts (apoptosed) |
| Antibody | Anti-heme oxygenase 1 (rabbit polyclonal) | Abcam | ab13243 | IF (1:150) |
| Antibody | Anti-8-oxoguanine (mouse monoclonal) | Abcam | ab206461 | IF (1:150) |
| Antibody | Anti-mouse F4/80 (rat monoclonal) | Santa Cruz Biotechnology | sc-52664 | IF (1:50) |
| Antibody | Anti-arginase-1 (goat polyclonal) | Abcam | ab60176 | IF (1:150) |
| Antibody | Anti-MERTK (rabbit polyclonal) | Abcam | ab95925 | IF (1:150) |
| Antibody | Anti-mouse iNOS (rabbit monoclonal) | Cell Signaling Technology | #13120 | IF (1:100) |
| Antibody | Anti-myeloperoxidase (rabbit polyclonal) | Abcam | ab9535 | IF (1:75) |
| Antibody | Anti-CD38 (rabbit monoclonal) | Cell Signaling Technology | #68336 | IF (1:75) |
| Antibody | Anti-myoglobin (rabbit monoclonal) | Cell Signaling Technology | #25919 | IF (1:100) WB (1:1000) |
| Antibody | Anti-GAPDH (mouse monoclonal) | Merck | MAB374 | WB (1:5000) |
| Antibody | Anti-mouse light chain (goat polyclonal) | Merck | AP200P | HRP conjugate WB (1:10,000) |
| Antibody | Anti-rabbit IgG (goat polyclonal) | Merck | AP132P | HRP conjugate (1:3000) |
| Antibody | Anti-mouse citH3 (rabbit polyclonal) | Abcam | ab5103 | IF (1:150) |
| Antibody | Anti-nitrotyrosine (mouse monoclonal) | Abcam | ab7048 | IF (1:100) |
| Antibody | Anti-hemoglobin (rabbit polyclonal) | Proteintech | 16665-1-AP | IF (1:100) |
| Antibody | Anti-hemopexin (rabbit polyclonal) | Proteintech | 15736-1-AP | IF (1:100) |
| Antibody | Anti-haptoglobin (rabbit polyclonal) | Proteintech | 14537-1-AP | IF (1:100) |
| Antibody | Anti-rabbit IgG (donkey polyclonal) | Abcam | ab150064 | Conjugated to Alexa Fluor 594 IF (1:1000) |
| Antibody | Anti-rabbit IgG (donkey polyclonal) | Abcam | ab150075 | Conjugated to Alexa Fluor 647 IF (1:1000) |

*Appendix 1 Continued on next page*

*Appendix 1 Continued*

| Reagent type (species) or resource | Designation | Source or reference | Identifiers | Additional information |
|---|---|---|---|---|
| Antibody | Anti-goat IgG (donkey polyclonal) | Abcam | ab150129 | Conjugated to Alexa Fluor 488 IF (1:1000) |
| Antibody | Anti-mouse IgG (donkey polyclonal) | Abcam | ab150107 | Conjugated to Alexa Fluor 647 IF (1:1000) |
| Sequence-based reagent | Myoglobin common reverse | IDT | PCR primers | CCTTGCTCTGACTGTGTTAGCCTCAG |
| Sequence-based reagent | Myoglobin wildtype forward | IDT | PCR primers | GTGTGACAGAGTGGCTGTCATACTTTGT |
| Sequence-based reagent | Myoglobin knockout forward | IDT | PCR primers | CCGTTGACCCACCTTGTCTCAA |
| Sequence-based reagent | Pax7 common forward | IDT | PCR primers | GCTGCTGTTGATTACCTGGC |
| Sequence-based reagent | Pax7 wildtype reverse | IDT | PCR primers | CTGCACTGAGACAGGACCG |
| Sequence-based reagent | Pax7 transgene reverse | IDT | PCR primers | CAAAAGACGGCAATATGGTG |
| Sequence-based reagent | Brainbow2.1 common reverse | IDT | PCR primers | CCAGATGACTACCTATCCTC |
| Sequence-based reagent | Brainbow2.1 wildtype forward | IDT | PCR primers | AAAGTCGCTCTGAGTTGTTAT |
| Sequence-based reagent | Brainbow2.1 transgene forward | IDT | PCR primers | GAATTAATTCCGGTATAACTTCG |
| Peptide, recombinant protein | Myoglobin (equine) | Sigma-Aldrich | M0630 | |
| Peptide, recombinant protein | Interferon gamma | Stem Cell Technologies | 78021.1 | |
| Commercial assay or kit | GenElute Mammalian Genomic DNA Miniprep Kits | Sigma-Aldrich | G1N70 | |
| Commercial assay or kit | Iron assay kit | Abcam | ab83366 | |
| Commercial assay or kit | Perls' Prussian blue iron stain | Abcam | ab150674 | |
| Commercial assay or kit | DeadEnd Fluorometric TUNEL System | Promega | G3250 | |
| Commercial assay or kit | Mouse Luminex Discovery Assay | R&D Systems | LXSAMSM-21 | |
| Commercial assay or kit | ROS-Glo H2O2 Assay | Promega | G8821 | |
| Commercial assay or kit | Efferocytosis Assay Kit | Cayman Chemical | 601770 | |
| Commercial assay or kit | MycoAlert PLUS mycoplasma detection kit | Lonza | LT07-705 | |
| Commercial assay or kit | FTA Sample Collection Kit for Mouse Cell Authentication Service | ATCC | 137-XV | |
| Chemical compound, drug | Cardiotoxin | Sigma-Aldrich | C9759 | *N. mossambica* venom |
| Chemical compound, drug | Naniproin | PMID:27173146 | | *N. nigricollis* venom |
| Chemical compound, drug | Deferoxamine mesylate | Sigma-Aldrich | D9533 | |
| Chemical compound, drug | Tamoxifen | Sigma-Aldrich | T5648 | |

*Appendix 1 Continued on next page*

*Appendix 1 Continued*

| Reagent type (species) or resource | Designation | Source or reference | Identifiers | Additional information |
|---|---|---|---|---|
| Chemical compound, drug | Buprenorphine (Bupredyne) | Jurox Animal Health | | |
| Chemical compound, drug | Isoflurane | Piramal Critical Care | | |
| Chemical compound, drug | Menadione | Sigma-Aldrich | M5625 | |
| Chemical compound, drug | Lipo-polysaccharide | Sigma-Aldrich | L4391 | |
| Chemical compound, drug | Sodium chloride solution, 0.9% | Sigma-Aldrich | S8776 | |
| Software, algorithm | GraphPad Prism | GraphPad Software | | |
| Software, algorithm | MATLAB | This paper | | Thresholding of blue color from Prussian blue iron deposits |
| Other | Ceramic magnets | Magnetic Source | CD14C | See 'Murine pressure injury model' |
| Other | Neodymium magnets | Liftontech Supreme Pte Ltd | | See 'Murine pressure injury model' |
| Other | Vectashield Hardset with DAPI | Vector Laboratories | H-1500-10 | See 'Immunofluorescence staining' |
| Other | BODIPY 581/591C11 probe | Invitrogen | D3861 | See 'BODIPY 581/591C11 detection' |
| Other | Eosin Y solution | Sigma-Aldrich | HT110116 | See 'Histopathology scoring of H&E-stained sections' |
| Other | Shandon Gill Hematoxylin 2 | Thermo Fisher | 6765007 | See 'Histopathology scoring of H&E-stained sections' |

