## [Editor Report]

It is known that muscle pressure injury (mPI), tissue damage caused by sustained pressure, is difficult to be healed although the mechanism underlying has been poorly understood. In this study, the authors demonstrated a convincing evidence that myoglobin (Mb) released at the site of injury plays an important role in the size, severity, oxidative damage, and poor healing of mPI by causing the induction of immune cell (in particular phagocyte) death and delaying the clearance of dead tissues using Mb KO mice and iron chelation by deferoxamine. The authors’ findings are valuable in developing a novel therapeutic option for mPI although further clinical corroboration of these findings would be of even greater value.

---

## [Decision Letter]

[Editors' note: this paper was reviewed by Review Commons.]

---

## [Author Response]

General StatementsIssue A. Removing the claim of phagocytic dysfunction.Reviewer #1:“The study is well written, clear, and the experiments are carefully presented and conducted. Although the text is usually very detailed and nicely referenced, some of the claims should be dampened. Notably the title since the phagocytic dysfunction is not evidenced by the results…Concerning immune cells, the conclusion cannot be that myoglobin impedes their phagocytic function. All the data concur that in the absence of myoglobin there are more immune cells in the regenerating muscle at day 3. Consequently, more macrophages will lead to a better cleansing of debris. Thus, the difference would not rely on phagocytic properties per se, but more on the number of macrophages that arrive at the site of injury.”

We clarify a major misunderstanding. All instances of the phrase “failure of phagocytosis” or **“**phagocytic dysfunction” should be re-worded as “persistence of dead tissue**”** or simply “necro-slough.” The word “phagocytic” contains an implication of “cells” or micro-scale issues, which was not our thinking. We apologize for the ambiguity. Our claim is strictly about the millimeter-scale tissue outcome, not about cell activities to cause the tissue outcome.

Manuscript revisions:

Changed the title to omit the term “phagocytic dysfunction”.Changed the text to emphasize tissue physiology and not evoke concepts of cell biology. Changed terminology so that all “failure of phagocytosis” will be written as “persistence of dead tissue.”Our abstract now says, “Unlike acute injuries (from cardiotoxin), mPI regenerated poorly with a lack of viable immune cells, *persistence of dead tissue (necro-slough)*, and abnormal deposition of iron.” (The old version had said, “mPI regenerated poorly with a lack of viable immune cells, *failure of phagocytosis*….”)Our hypothesis statement now reads as follows: “Given that pressure ulcers often have slough or eschar, we hypothesize that mPI will have *persistence of dead tissue* in the wound bed, and that sterile mPI will have slough, despite the absence of bacterial biofilm.” This is a clinically oriented claim about the relationship between bacterial infection and sloughing, not a cell biology claim about the relationship between macrophages and efferocytosis.

Issue B. Macrophage characterisationReviewer #1:“It would be interesting to see if myoglobin prevents monocyte or macrophage migration/chemotaxis.”

This is an interesting question. We performed in vitro testing of RAW264.7 cell migration/chemotaxis using serum differential in a trans-well migration assay. Results are shown in Author response image 1. Because this assay is yielding large error bars without affecting our argument (positively or negatively), we prefer to de-emphasise the issue of monocyte/macrophage migration, and instead emphasise the issue of vasculature (see Issue D below).

**Author response image 1. sa2fig1:** Chemotaxis of macrophages measured using a Transwell migration assay. RAW 264.7 monocyte cells were incubated with pro-inflammatory stimuli (10ng/ml IFNγ + 100ng/ml LPS), for 24h prior to 24h treatment with varying concentrations of myoglobin. Macrophages (1×10^4^ cells/well in DMEM without FBS) were loaded into the upper portion of a 6.5-mm Transwell with 8.0-µm Pore Polycarbonate Membrane Insert (Corning, NY, USA). DMEM with 10% FBS supplementation was placed in the lower chamber. The cells in the bottom chamber and bottom of the insert (i.e., the migrated cells) were collected and quantified using the Cell Titre-Glo assay, with luminescence measured using a microplate reader. *n* = 3-4 replicates. CFM = cell-free media. Mean ± standard deviation. “ns” not significant, **<0.01 Statistical significance was computed by one-way analysis of variance (ANOVA) with Tukey post-hoc test.

Issue C. Macrophage phagocytosis / efferocytosisReviewer #2:“7. Macrophage phenotype (inflammatory vs. anti-inflammatory/reparative) can be achieved by RTqPCR, using well-define combination of mRNA encoding proteins associated with inflammatory (e.g. iNos, Cox-2, Cd86) or anti-inflammatory (Ym1, Arg-1, RELMa, Cd206).8. The in vitro experiments with macrophages could be further supported by in vivo experiments, where types of injury (mPI vs. CTX) and mouse genotypes (Mb-/- vs. WT) could be evaluated for the ability of macrophages to perform efferocytosis: coupling apoptotic cell detection (Tunel staining) to macrophage immunostaining. Quantification of overlapping signal would give some information (albeit indirect) regarding the macrophages' ability to clear the tissue from dead cells. From my perspective, this would be the minimal set of data required to highlight a potential "efferocytic failure" in mPI.”Reviewer #1 cross-comment:“Point 8 should be investigated if the authors wish to claim about efferocytosis.”Reviewer #2 cross-comment:“I would rather suggest the authors to focus on strengthening the description of their model of injury (mPI) versus CTX, on key points that are known to influence tissue repair/regeneration as well as their main findings: evaluation of the injur’'s surface area, myoglobin deposition/accumulation in the wound, better describing the immune response ensuing the injury. Beyond these points, the authors can then re-assess whether or not to include the role of myoglobin on monocyte/macrophage infiltration on the site of injury and the phagocytic activity of these recruited macrophages as part of this manuscript.”

We did not intend to make a claim about efferocytosis or phagocytosis of monocytes/macrophages. We have amended the title and text to omit “phagocyte dysfunction”, as mentioned in Issue A.

That being said, both reviewers requested additional study of macrophages, and provided various options for how to address the underlying question of what macrophages in mPI are doing. Reviewer #2 recommended functional assays of phagocytic capacity in vitro, so we assessed the impact of myoglobin on M1-differentiated RAW264.7 macrophages using an efferocytosis assay. Results showed that Mb treatment caused decreased efferocytosis in vitro, which is now reported in Figure 4—figure supplement 2A. Note that we maximised cell viability during this Mb treatment assay by using an M1 polarisation protocol [1] with a lower dose of IFNγ (10 ng/ml instead of 20 ng/ml) and shorter duration (see Supplementary Methods). We have placed this data in the Supplement (and do not mention it in the Discussion) because we believe impaired efferocytosis could be a side-effect of cell stress, and not necessarily a specific effect of Mb. Mb treatment induced ROS in a dose-dependent manner (Figure 4—figure supplement 2B).

In tissue sections ex vivo, TUNEL staining (for apoptotic debris) was far higher in saline-treated mPI than in CTX or DFO-treated mPI (Figure 5—figure supplement 1), but we were unable to find co-localization of F4/80 and TUNEL staining in our samples, so we could not measure increased or decreased co-localization (as a readout of macrophage engulfment of debris). The increased TUNEL staining might have arisen from increased production (e.g., secondary progression of the wound) or from decreased uptake (e.g., impaired efferocytosis).

Manuscript revisions:

Removed the poor wording that suggested a claim about macrophage phagocytosisNew data in Figure 4—figure supplement 2New data in Figure 5—figure supplement 1

Issue D. VasculatureReviewer #1:“Another aspect is how cells reach the injured area. In the Discussion section, there is reference to a SupplFig6, which seems to be not the good one in the document. In the FigS6 described in the text, it is mentioned that cells are kind of "stopped" at the boundaries of the damage. This is very interesting. If the vessels are physically flattened or squashed by the injury, extravasation can not occur properly, in comparison with cardiotoxin. Then the role of myoglobin in extravasation, or in the "reshaping" of the vessels after the removal of the magnet would be interesting to investigate. Of note, in the myoglobin deficient mice, the vascular network is increased, favoring immune cell infiltration.Given that the study is already quite huge with numerous experiments, the reviewer is reluctant to ask for additional experiments… Also the points mentioned above, about the role of myoglobin in immune cell infiltration and the role of myoglobin in the vessel properties should be at least discussed, if they are not experimentally addressed.– Discussion end of page 10. Unfortunately, suplFig6 is missing (and is not called in the result section).”

We apologise for the omission of SupplFig6 from the journal version. The previous version of Suppl. Figure 6 (in bioRxiv) showed side-by-side comparisons of young and old saline-treated mPI, and it has been included in the current submission (renumbered as Figure 3—figure supplement 2). We also provide additional data (Figure 2—figure supplement 1) showing cells reaching, or failing to reach, the injured area at day 3 post-injury. Specifically, it shows that the uncompressed periphery of mPI wounds have intact immune cells, while the compressed region lacks viable immune infiltrate (Figure 2—figure supplement 1).

In addition, we provide new data showing damaged blood vessels in Mb-wildtype mPI compared to intact vasculature in CTX at day 3 post-injury (Figure 1—figure supplement 2). Damaged vasculature has a number of important implications, which are now elaborated in the Discussion at lines 371-377, as follows: “Pressure has been reported to cause disruption to arterial supply, venous clearance, and lymphatic drainage (4, 51, 52), although not always (79). Lymph vessels are occluded at very low pressure, and impaired lymphatic flow can persist after pressure is removed (80). Impaired circulation and/or impaired drainage could cause waste factors such as myoglobin or iron to accumulate in the mPI, and could impair the influx of iron detoxification factors. Poor drainage also occurs in chronic ulcers of venous insufficiency, lymphedema, and other vascular diseases. Toxicity from globin iron might be a shared pathology among these wounds.”

Manuscript revisions:

New data in Figure 1—figure supplement 2New data in Figure 2—figure supplement 1Next text about damaged vasculature (Results section, lines 120-123, Discussion section, lines 371-377)Figure 3—figure supplement 2 (previously Suppl. Figure 6 – comparison of young and old saline-treated mPI) is now mentioned in the Results section

Issue E. Length of DiscussionReviewer #1:“Also the text is very long, notably the discussion, which contains 18 sections (!) and several sections are not very informative and poorly referenced, looking more as a thesis dissertation than as an article discussion… Given that the study is already quite huge with numerous experiments, the reviewer is reluctant to ask for additional experiments. Rather, the reviewer suggests to reshape the text, remove unnecessary details to get straight to the points and to emphasize the important results”“- Discussion sections about oxidative stress, endogenous iron, prevention studies, antiDAMPs strategies, slough and debridement are poorly informative and poorly referenced and should be either removed or shortened”.

We have shortened the discussion and conclusion by 33%, especially the sections entitled “the context of oxidative stress”, “anti-DAMP strategies”, and “debridement of slough.” (These changes were submitted in our December 2022 proposed revision and have not been marked-up as changes in the current submission.)

Issue F. Comparing mPI and CTX woundsReviewer #2:“The authors hypothesis regarding the role of Mb in the pathophysiology of ulcerative pressure injuries is interesting. However, the work here seems quite preliminary with major points remaining to be clarified before considering reaching the author's conclusion. I find the direct comparison made between the two types of injury, CTX and mPI, difficult to interpret. From my perspective, a more rigorous and systematic comparison between the two models of injury would be key to convincingly convey the findings of this work, especially regarding key features impacting repair. My major comments are listed below.1. Time for tissue repair not only depends on the type of injury, but also on the extent of the injury. In other words, how the mPI and CTX models compare in terms of surface of injured tissue (and resulting ischemia)”

We interpret the feedback to mean that the surface area of injured muscle should be comparable between CTX and mPI, in order to claim that the tissue repair is different between the two injuries. We now provide measurements showing that both CTX and mPI have comparable size of dead muscle at the initial timepoint (Supplementary File 1a).

However, we caution readers that different types of wounds are not truly comparable. We have provided the contrast between CTX and mPI for hypothesis-generation, not as a controlled experiment for hypothesis-testing. To clarify the limits of using CTX as a reference for understanding mPI, we provide additional data mentioned in Issue D showing dissimilarity between CTX and mPI in vasculature and thickness.

Manuscript revisions:

New data in Supplementary File 1aNew data in Figure 1—figure supplement 2We delete “normal” from our description of CTX regeneration, and better explain the purpose of comparing CTX and mPI.

Issue G. Accumulation and detoxification of globin proteins in the woundsReviewer #2:“2. Is myoglobin also released in the injured tissue after CTX and how does it compare to mPI (Mb+ surface area, quantity)?9. What about expression levels of proteins involved in heme/iron detoxifying proteins haptoglobin and hemopexin? Are they present in the injured tissue and are they differentially expressed between types of injury and mouse genotypes? Same goes for their receptors (CD163 and CD91, respectively): are they differentially expressed on macrophages found in the injured tissue.– Is hemoglobin found mPI and CTX wounds from WT or Mb-/- mice?”

We interpret this feedback to mean that measurements of myoglobin and iron detoxification factors would help explain why mPI has higher iron and what consequences the iron may have. We agree this might be interesting, provided we also bear in mind the destruction of vessels in mPI (described in Figure 1—figure supplement 2 and Issue D above). Damaged vasculature could explain why iron-containing waste accumulates in mPI, while the same waste gets removed in CTX. To assess levels of globin proteins in the wound, we performed IF for myoglobin and hemoglobin in mPI at day 3 post-injury. Results showed higher levels of myoglobin and hemoglobin in mPI than in CTX (Figure 2—figure supplement 2A-H and Figure 2—figure supplement 2Q-R, for *n* = 3, with all wounds saline-treated).

To assess iron detoxification factors in the wound, we performed IF for haptoglobin and hemopexin (Figure 2—figure supplement 2I-P and Figure 2—figure supplement 2S-T), which were very low, relative to our positive controls (liver and spleen, not shown). We found no significant difference between mPI and CTX at day 3 post-injury. Possible interpretations of this finding are that day 3 might be too late to see the peak of iron detoxification for the CTX injury, and the damaged vasculature of mPI might decrease the influx of iron detoxification factors into high-iron mPI wounds.

Manuscript revisions:

New data in Figure 2—figure supplement 2A-H and Figure 2—figure supplement 2Q-RNew data in Figure 2—figure supplement 2I-P and Figure 2—figure supplement 2S-TNext text about damaged vasculature (Results section, lines 120-123, Discussion section, lines 371-377)

Issue H. Additional mouse modelsReviewer #2:“3. Does Mb co-injection with CTX mimics mPI injury in terms of inflammation and repair kinetics?”Reviewer #2 cross-comment:“I would rather suggest the authors to focus on strengthening the description of their model of injury (mPI) versus CTX, on key points that are known to influence tissue repair/regeneration as well as their main findings: evaluation of the injury's surface area, myoglobin deposition/accumulation in the wound, better describing the immune response ensuing the injury.”

Reviewers did not prioritize an additional mouse injury model, and our Dec 2022 revision plan offered other experiments instead.

Reviewer #2:“4. Does in situ Mb supplementation in Mb-/- mice worsens mPI repair to an extent that is comparable to WT mice?5. A better characterization of the inflammatory between types of injury and mice (Mb-/- vs. WT) before and after 3- and 10-days post-injury would be very informative. Comparing the relative proportions of leukocyte populations would provide valuable information regarding the kinetics of the repair process.”

Reviewers did not prioritize further study of knockout mice (Mb-/-), and our Dec 2022 revision plan offered other experiments instead.

Issue I. Characterising the immune landscape in mPIReviewer #2:“6. Macrophages in particular play a central role in the orchestration of tissue repair, through their immunomodulation abilities. On the same token, characterizing macrophage infiltrates (number per surface area of injured tissue) and phenotype would potentially provide valuable information to link observed differences between types of injuries to Mb. Ideally, assessment of leukocyte and macrophages infiltration and populations would be analyzed by flow cytometry after injured (vs. uninjured) tissue dissociation (enzymatic or mechanical). Otherwise, although less quantitative, this can also be done by cell infiltration using specific immunostaining and quantification (cell number/injured tissue surface area).

To assess leukocytes in mPI wounds, we performed immunostaining for CD38 and myeloperoxidase at day 7 post-injury. CD38 is a marker of CD4^+^, CD8^+^, B and Natural Killer cells, and myeloperoxidase is a marker of neutrophil extracellular traps (NETs/NETosis). Immunostaining (Figure 6—figure supplement 3A-G) showed that myeloperoxidase staining was high in mPI+Saline, and lower in DFO-treated. The lower level of myeloperoxidase staining in DFO-treated (day 7) resembled the level in CTX injury at day 3 (*n* = 3 replicates each). Immunostaining for CD38 (Figure 6—figure supplement 3H-N) showed similar expression levels between mPI+Saline at day 7 and CTX+Saline (CTX) at day 3, and increased expression in mPI+DFO at day 7, compared to CTX (*n* = 3 replicates each). These findings are now mentioned in the Results section alongside CitH3 and MerTK immuno-staining, but are not mentioned in the discussion. As some of our staining may not localize to cells but to cell debris and extracellular molecules, we chose not to quantify fluorescence based on cell number per tissue area. Instead, we provide quantification using fluorescence thresholding [2] of images of injured sections, and report the average of the injured frames for each wound/mouse.

We also performed immuno-staining with additional antibodies that failed quality-control checks; these were for Ly6G, a marker of neutrophils, and CD86, a marker of dendritic cells, macrophages, B cells and other antigen-presenting cells.

For the broader goal of characterising immune response in mPIs, we did already measure many cytokines and general immune-related factors at different timepoints. Surprisingly few of the analytes had any significance change upon Mb knock-out or DFO treatment (Supplementary Files 1d, 1f and 1g), despite the large fold-changes seen in damage-related epitopes of oxidative stress. We wish to avoid making any claims about the immune system in mPI, other than the absence of intact immune cells in untreated day-3 mPI wounds, and the DFO-induced increase in the presence/influx of immune cells at days 7 & 10. These serendipitous findings about immune cells are not required for any of the five hypotheses listed in our introduction.

Manuscript revisions:

New data in Figure 6—figure supplement 3

Issue J. Minor commentsReviewer #1:“- Page 5: "The panniculus layer of mPI was nearly devoid of intact immune cells." It is not clear here if the authors refer to the absence of immune cells or to damage of immune cells present in the injured area.”

We refer death and damage of the immune cells in the injured area, rather than complete absence of these cells. This is discussed in lines 384-396.

“- Page 6: "In summary, measures of innate immune response became less abnormal after Mb knockout". Cardiotoxin injury is not the "normal" situation since this kind of injury is not physiological and is highly inflammatory as compared with others (Hardy et al., Plos One 2016).”

Thank you. We have omitted the word “normal” in our description of acute cardiotoxin injury.

Reviewer #2:“- Figure 2: HO-1 staining seem decreased in mPI compared to CTX and thus doesn't support the quantifications. Please 2x-check quantifications and images to provide consistent quantifications-illustrations pairing.”

Thank you. We have doublechecked the quantifications-illustrations pairing and replaced Figure 2L and Figure 5E with more representative images. Primary images, data and quantification spreadsheets have been uploaded to Zenodo at doi:10.5281/zenodo.7750287. The Data Availability section containing the Zenodo link has been moved from Supplementary Methods to Methods (in the main text).

“– Figure 5 C, E, G: please provide illustrations for control treatment.– Figure 6 K, L, M: please provide illustrations for control treatment.”

The controls for Figure 5C, 5E, and 5G were already shown in Figure 2 (2I, 2L and 2B), and we hesitate to show them again without permission to duplicate from the editor. Similarly, controls for Figure 6K, 6L and 6M are already shown in Figure 2F-G.

“– Figure 5J: it would have been nice to add Mb-/- mice to the comparison.”

We have added Mb-/- mice to the comparison of external wound area, up to the day 3 timepoint.

“– Figure 8: please maintain consistency in the way you convey data between timepoints: area of regenerated (E, F) or unregenerated (G) tissue.”

For consistency, we have moved Figure 8G to Figure 9 (Figure 9K) such that Figure 8 shows regeneration at the 40-day timepoint and Figure 9 shows regeneration and fiber morphology at the 90-day timepoint.

“– Bodipy is not a probe for lipid peroxidation. Due to its lipophilic nature, this dye can be used as a generic lipid satin to image intracellular lipid depots. Therefor the experiments using bodipy as a proxy for lipid peroxidation is incorrect and derived conclusions erroneous. Modulation in bodipy signals probably reflects modulation of intracellular lipid deposition.”

Clarifying text and citation on the BODIPY 581/591 fluorescent probe that we used (Invitrogen, MA, US) have been added in the Results section.

“– Provide concentrations and treatment times in figure legends (sup Figure 3).”

Thank you, we have added the concentrations and treatment times to the figure legend. Further details are in the Methods.

“– Show all the data mentioned in the manuscript (DNA gel electrophoresis supp Figure 4)”

We have added the gel electrophoresis results to Figure 3—figure supplement 1 (previously Suppl. Figure 4).

“– Indicate the number of experimental repeats and the statistical tests used in the figure legends.”

We have added the *n* numbers, statistical tests, and definitions of centre to the figure legend. We have edited the main text to report exact p-values instead of the p-value range (e.g., p = 0.0251 instead of p < 0.05).

“– Missing information in supp Figure 5 A-D: which images from WT or Myb-KO?”

Thank you. We have added labels for Figure 4—figure supplement 1 (previously Suppl. Figure 5), Figure 6—figure supplement 1 and Figure 7—figure supplement 1.

Issue K. SignificanceReviewer #1 (Significance):“This study is of interest because it provides insights on muscle regeneration after an injury that may occur in daily life just as contusion, or crush, to the contrary of the whole muscle necrosis induced by cardiotoxin (the main model used in the field). In that aspect, it is more physiological and of interest for the readers in the fields of in muscle regeneration and tissue trauma. The study is descriptive, but is very well conducted and well discussed. Additional experiments investigating the impact of myoglobin on vessel properties/extravasation of immune cells would raise the impact of the study but the study is publishable as it is (with editing as suggest above). The field of expertise of the reviewer is muscle regeneration and inflammation.”Reviewer #2 (Significance):“The field of tissue repair and regeneration is an exciting field and improving our understanding of the molecular mechanisms involved in muscle tissue injury has clear and impactful clinical applications. The pathophysiological mechanism involving Mb that the author address in this work has the potential to interest both basic science and clinical researchers and can potentially benefit not only the field of skeletal muscle regeneration but also the field of cardiac remodeling.”

We are very grateful to the reviewers for their insightful and highly constructive feedback.

Issue L. Minor edits made by the authors

The section, “Data availability”, which discloses the link to the Zenodo database containing the mice numbers and primary data has been moved from Suppl. Methods (Suppl. Text) to Methods in the main text. The Zenodo description and DOI have been updated to reflect the changes made in this revision.Suppl. Figure 13A-D have a different image set to show F4/80, CitH3 and DAPI triple-stain in saline-treated mPI (day 3). DAPI staining was not shown previously (only F4/80 and CitH3).Suppl. Figure 15E has a blue dashed line added to the graph for the level of MerTK fluorescence in uninjured skinfold.Exact p values are now reported in the main text, as requested by the editorial office.Statistical tests used, exact values of *n*, definitions of center, and methods of multiple test correction methods have been included in the figure captions, as requested by the editorial office.The methods for TUNEL staining, cell culture conditions and treatments and in vitro assays have been added in the Supplementary Methods.Cell authentication of the cell lines by STR profiling has been provided, as requested by the editorial office.The mycoplasma status of the cell lines has been included, as requested by the editorial office.A Key Resources Table has been provided, as requested by the editorial office.The Supplementary Figures and Supplementary Tables have been separated for better readability.The acknowledgements section has been updated.

References

[1] Cao Q, Yao J, Li H, Tao B, Cai Y, Xiao P, Cheng H, Ke Y. (2017). Cellular phenotypic analysis of macrophage activation unveils kinetic responses of agents targeting phosphorylation. *SLAS Discov*, 22(1):51-57.

[2] Shihan MH, Novo SG, Le Marchand SJ, Wang Y, Duncan MK. (2021). A simple method for quantitating confocal fluorescent images. *Biochemistry and Biophysics Reports*, 25:100916.